# Similarities and differences in Alzheimer's dementia comorbidities in racialized populations identified from electronic medical records

Sarah R. Woldemariam [1], Alice S. Tang [1,2], Tomiko T. Oskotsky [1,3], Kristine Yaffe[4] & Marina Sirota [1,3]✉

## Abstract

**Background** Alzheimer's dementia (AD) is a neurodegenerative disease that is disproportionately prevalent in racially marginalized individuals. However, due to research underrepresentation, the spectrum of AD-associated comorbidities that increase AD risk or suggest AD treatment disparities in these individuals is not completely understood. We leveraged electronic medical records (EMR) to explore AD-associated comorbidities and disease networks in racialized individuals identified as Asian, Non-Latine Black, Latine, or Non-Latine White.

**Methods** We performed low-dimensional embedding, differential analysis, and disease network-based analyses of 5664 patients with AD and 11,328 demographically matched controls across two EMR systems and five medical centers, with equal representation of Asian-, Non-Latine Black-, Latine-, and Non-Latine White-identified individuals. For low-dimensional embedding and disease network comparisons, Mann-Whitney $U$ tests or Kruskal-Wallis tests followed by Dunn's tests were used to compare categories. Fisher's exact or chi-squared tests were used for differential analysis. Spearman's rank correlation coefficients were used to compare results between the two EMR systems.

**Results** Here we show that primarily established AD-associated comorbidities, such as essential hypertension and major depressive disorder, are generally similar across racialized populations. However, a few comorbidities, including respiratory diseases, may be significantly associated with AD in Black- and Latine- identified individuals.

**Conclusions** Our study revealed similarities and differences in AD-associated comorbidities and disease networks between racialized populations. Our approach could be a starting point for hypothesis-driven studies that can further explore the relationship between these comorbidities and AD in racialized populations, potentially identifying interventions that can reduce AD health disparities.

### Plain language summary

Black- and Latine- identified individuals in the United States are more likely to have Alzheimer's dementia (AD) relative to Asian- and White-identified individuals. Despite this, Black- and Latine- identified individuals are less likely to be included in studies that attempt to understand and treat AD. Patients' medical information, electronically recorded by healthcare providers, was used to explore whether patients with AD were more likely to have different conditions relative to patients who do not have AD. We did this analysis separately for Asian-, Non-Latine Black-, Latine- and Non-Latine White- identified individuals for a total of four analyses. While we found many conditions that were shared by all individuals, a few, such as lung-related diseases, may be more common in specific identified race and ethnicity categories.

[1] Bakar Computational Health Sciences Institute, University of California San Francisco, San Francisco, California, USA. [2] School of Medicine, University of California San Francisco, San Francisco, California, USA. [3] Department of Pediatrics, University of California San Francisco, San Francisco, California, USA. [4] Department of Psychiatry and Behavioral Sciences, University of California San Francisco, San Francisco, California, USA. ✉email: Marina.Sirota@ucsf.edu

Alzheimer's dementia (AD), which comprises 60–80% of all dementia cases, is a currently incurable heterogeneous neurodegenerative disease characterized by extracellular β-amyloid plaques and intracellular phosphorylated tau neurofibrillary tangles[1]. AD generates substantial personal and economic costs, with the number of individuals 65 years of age and older in the U.S. diagnosed with AD expected to nearly double to 12.7 million by 2050. AD is also disproportionately prevalent in racially marginalized individuals in the U.S.; specifically, AD has a higher prevalence in Black- and Latine- relative to Asian- and White- identified individuals (we use Latine here as a gender-neutral term that is more intuitive to pronounce in Spanish and other Romance languages). The factors underlying this are not fully understood but are likely attributable to social determinants of health (SDoH) and the closely related effects of racism, which drive disease disparities generally[2,3]. For instance, cardiovascular risk factors like hypertension and diabetes, which also increase dementia risk, are often exacerbated in racially marginalized individuals through exposure to racism[4,5].

Over the past decade, there has been a rapid rise in electronic medical record (EMR) implementation across U.S. health centers[6]. This wealth of clinical data can be utilized to characterize complex and heterogeneous conditions like AD from many individuals and institutions. Indeed, EMR data have been leveraged to perform sex-stratified deep phenotyping of AD, including exploration of comorbidities, medications, and lab values that may be differentially associated in individuals with AD depending on their identified sex[7]. In another study, EMR-derived disease networks in racialized populations revealed differences in disease trajectories, demonstrating how network-based analyses can identify differences in health outcomes between racialized populations (we use the term racialized to emphasize the process by which people are assigned racial categories; while definitions of 'racialized' often exclude White-identified individuals, they are included in our usage)[8,9]. Prior epidemiological, hypothesis-driven, and cross-sectional studies have explored differences in the prevalence of specific AD- and dementia-associated comorbidities between racialized individuals, including hallucinations, sleep disturbances, dementia-related behavioral symptoms, mental disorders, diabetes, and cardiovascular comorbidities[10–13]. While a big data approach to analyzing EMR has the potential to address health disparities in diseases like AD, especially if demographic characteristics, geographic location, and SDoH are systematically collected, there have been no studies using this approach to identify which comorbidities in the EMR are statistically significant in patients with AD, stratified by identified race and ethnicity (R&E)[14].

Here, we perform a case-control observational study to identify AD-associated comorbidities in Asian-, Non-Latine Black-, Latine-, and Non-Latine White- identified individuals who received care at the University of California San Francisco (UCSF), with the validation cohort comprising Asian-, Non-Latine Black-, Latine-, and Non-Latine White- identified individuals who received care from either UC Davis (UCD), UC Irvine (UCI), UC Los Angeles (UCLA), or UC San Diego (UCSD). We also perform low-dimensional embedding as well as generate and compare AD and control disease networks, stratified by identified R&E. Together, these analyses uncover EMR phenotypic profiles and networks that vary based on AD status and identified R&E, in addition to similar and different AD-associated comorbidities in racialized populations. Shared AD-associated comorbidities have been primarily established before and include essential hypertension, major depressive disorder, and urinary tract infection. Other less established AD-associated comorbidities are also found in specific racialized populations, such as respiratory diseases in Black- and Latine- identified individuals. The data-driven approaches we leverage here could generate hypotheses that could be tested to identify effective interventions to mitigate disparities in AD diagnosis and treatment.

## Methods

Analyses of UCSF and University of California Data Discovery Platform (UCDDP) de-identified EMR data were performed by UCSF employees under Institutional Review Board (IRB) approvals from UCSF, UCD, UCI, UCLA, and UCSD. Since only de-identified data were analyzed, the IRBs waived the need for written informed consent from patients for this study.

**Patient cohort selection**. Patients with AD were identified from the UCSF Observational Medical Outcomes Partnership (OMOP)-based EMR database in September and October 2021, which contained clinical data from over 5 million patients from January 1, 1982 to January 7, 2021. UCSF patients with AD were identified by inclusion criteria of having an estimated age of 65 years and older with at least one of the following ICD-10-CM codes for AD: G30.1, G30.8, or G30.9 (OMOP concept ids 35207357, 35207358, and 35207359, respectively). For de-identification purposes, dates were shifted by at most a year, and all patients over 90 years old have birth dates shifted to represent an estimated age of 90 years old in 2021. We used propensity score (PS) matching (MatchIt R package)[15] to identify control patients by estimating PS with a logistic regression model. Control patients were matched to patients with AD on sex, estimated age, identified R&E, and death status using a nearest neighbors method and a 1:2 AD:control ratio. Patients were matched by identified sex to remove potential differences in comorbidities amongst racialized populations due to differences in sex proportion, as previous studies have shown sex-specific AD-associated comorbidities[7]. Additionally, patients were matched on death status to account for patients who may have or may develop AD, but were deceased before a diagnosis could be made. To stratify analyses by identified R&E, subcohorts with equal numbers of patients identified as Asian, Non-Latine Black, Latine, and Non-Latine White (maintaining a 1:2 AD:control ratio) were identified after additional rounds of PS matching that matched patients stratified by identified R&E on sex, estimated age, and death status. A UCSF algorithm that uses self-identified R&E as inputs was primarily used to determine identified R&E[16]. A modified version of the UCSF algorithm for R&E categorization was used to identify R&E for the small number of patients that had an identified race and an identified ethnicity, but did not have a combined identified R&E. This version uses descriptions of patients' identified race and identified ethnicity in the database as inputs to determine patients' combined identified R&E.

**Mapping ICD diagnoses to phecodes**. Patients' ICD-9-CM and ICD-10-CM diagnoses, which include diagnoses before and after AD diagnosis for patients with AD, were pulled from the de-identified UCSF OMOP-based EMR database. Codes included those that were between approximately 6.8 years and 8.4 years before an AD diagnosis for patients with AD. These diagnoses were mapped to phecodes using the Phecode Map 1.2 with ICD-9 Codes[17,18] and Phecode Map 1.2 with ICD-10 Codes tables[19]. Phecodes that did not correspond to a phecode category (corresponding to the Excl. Phenotypes column in both Phecode Map 1.2 tables) were not included in analyses.

**Dimensionality reduction patient visualization**. Dimensionality reduction patient visualization was performed using the approach specified by Tang et al.[7] Patients were represented by one-hot-encoding of phenotype names with the exclusion of the AD

phenotype, then visualized in two dimensions using Uniform Manifold Approximation and Projection (UMAP) via the Python package umap-learn. Distributions of UMAP components were separately compared using two-sided Mann-Whitney $U$ tests (for two comparisons) or Kruskal-Wallis tests followed by post hoc two-sided Dunn's tests (for more than two comparisons) based on patients' AD status and identified R&E.

**Identified R&E-stratified AD vs. control differential analysis of comorbidities**. Identified R&E-stratified AD vs. control differential analysis of comorbidities, represented by phenotypes, were separately performed for Asian-, Non-Latine Black-, Latine-, and Non-Latine White-identified patients at UCSF. For each analysis, patients with AD were separately compared with matched identified R&E controls using two-sided Fisher's exact (if <5 patients in any category) or chi-squared tests to identify significant phenotypes among each racialized population, with the significance threshold defined as a Bonferroni-corrected $p$-value < 0.05. Results were visualized using an upset plot of overlapping significant phenotypes between populations and Manhattan plots that show phenotypes' $-\log_{10}$p-values grouped by phecode category for each racialized population.

**Identified R&E-stratified network analyses**. Identified R&E-stratified disease networks were created by using phenotypes shared by > 5% of patients (nodes) and paired phenotypes shared by > 5% of patients (edges). Separate networks were created for patients with AD and control patients, resulting in 8 networks. Network analysis was performed using the Cytoscape app Network Analyzer[20]. Metrics were compared between AD networks and between control networks separately using Kruskal-Wallis tests, followed by post hoc two-sided Dunn's tests. Metrics were also compared between identified R&E-stratified AD networks with respective controls using two-sided Mann-Whitney $U$ tests. A threshold of phenotypes shared by > 25% of patients (nodes) was used for network visualization.

**Validation in UC-wide EMR**. The de-identified UCDDP OMOP-based EMR database, which at the time of querying included clinical data from over 7 million patients from January 1, 2012 to July 31, 2021, was used to select UC-wide (i.e., UCD, UCI, UCLA, and UCSD) patients. In September 2021, patients with AD were identified by inclusion criteria of having an estimated age of 65 years and older with at least one of the following OMOP concept ids for AD that were mapped from the ICD-10-CM codes G30.1, G30.8, and G30.9: 4220313 (corresponding to SNOMED code 416975007, mapped from G30.1) and 378419 (corresponding to SNOMED code 26929004, mapped from G30.8 and G30.9). Different OMOP concept ids were used because, at the time of querying, the UCSF OMOP-based EMR database used non-OMOP-standard ICD-9-CM and ICD-10-CM diagnoses instead of OMOP-standard SNOMED diagnoses. For PS matching, UC-wide patients were additionally matched on UC location, and common support (to restrict matching of patients with AD and control patients to those with overlapping propensity score densities between the two groups) was utilized for the self- or provider-identified R&E-stratified PS matching. The modified version of the UCSF algorithm for R&E categorization was applied to the UC-wide validation cohort.

All analyses performed with UCSF patients were also performed for the UC-wide validation cohort. All ICD-9-CM and ICD-10-CM codes for patients were included, including codes between approximately 2.1 years and 2.3 years before an AD diagnosis for patients with AD. For dimensionality reduction, UMAP components were also compared based on patients' UC

location for the UC-wide validation cohort. Significant phenotypes at UCSF were mapped to the UC-wide validation cohort. To compare and evaluate UC-wide validation results, two-sided Spearman's rank correlation coefficients were calculated for the odds ratios of the mapped significant phenotypes between UCSF and UC-wide for each racialized population, with a significance threshold of $p$-value < 0.05; results were visualized using log-log plots. To compare network metrics between UCSF and UC-wide, metrics were each standard normalized to the metric (8 networks each at UCSF and UC-wide). Two-sided Spearman's rank correlation coefficients were calculated for the standard normalized metrics between UCSF and UC-wide, with a significance threshold of $p$-value < 0.05.

**Statistics and reproducibility**. Queries and mapping ICD diagnoses to phecode-corresponding phenotypes were both performed using SQL. Matching was performed using the MatchIt package in R. Low-dimensional embedding was performed using the Python package umap-learn. Statistical analyses were conducted using Python 3.8 and R. Statistical tests, including Fisher's exact, chi-squared, Kruskal-Wallis, Mann-Whitney $U$, and Spearman's rank correlation coefficient tests were primarily performed using SciPy. Two-sided Dunn's tests were performed using the R package dunn.test. Data were preprocessed using the Python package pandas. Visualizations were primarily created from the Python packages seaborn and matplotlib. Cytoscape, the Network Analyzer Cytoscape App, and the Python package py4cytoscape were used for network analysis and visualization. 7409 patients with AD, as well as 14,818 control patients, were identified in the UCSF cohort; therefore, we suggest that a similar number of patients be included for reproducibility.

**Reporting summary**. Further information on research design is available in the Nature Portfolio Reporting Summary linked to this article.

## Results

From the UCSF OMOP-based EMR database (>5 million patients), we identified 7409 patients with AD (Supplementary Fig. 1) and 14,818 PS-matched control patients. After a second round of PS-matching to obtain subcohorts of racialized populations, we identified 1688 patients with AD (422 patients for each identified R&E, mean estimated age 87.3 years old (6.0 standard deviation (SD))) and 3376 PS-matched controls (844 patients for each identified R&E, mean age 87.3 years old (6.0 SD)). From the UCDDP OMOP-based EMR database (>7 million patients), we identified 19,686 patients with AD (Supplementary Fig. 2) and 39,372 matched controls. After a second round of PS-matching to obtain subcohorts of racialized populations, we identified 3976 patients with AD (994 patients for each identified R&E, mean age 84.3 (6.0 SD)) and 7952 PS-matched control patients (1988 patients for each identified R&E, mean age 84.2 years old (6.0 SD)). We selected and analyzed patients who were primarily self-identified (UCSF) or were identified (in the UC-wide validation cohort) as Asian, Non-Latine Black, Latine, and Non-Latine White (Fig. 1). For the UC-wide analysis, only patients who received care from UCD, UCI, UCLA, and UCSD were included. A summary of demographic characteristics for patients at UCSF and UC-wide are shown in Table 1.

**Low-dimensional embeddings of patients' phenotypic profiles reveal separation based on AD status, R&E categories, and UC location**. We performed low-dimensional embedding of patients using non-AD ICD codes aggregated into phecode-corresponding phenotypes (referred to as phenotypes hereafter) to visualize

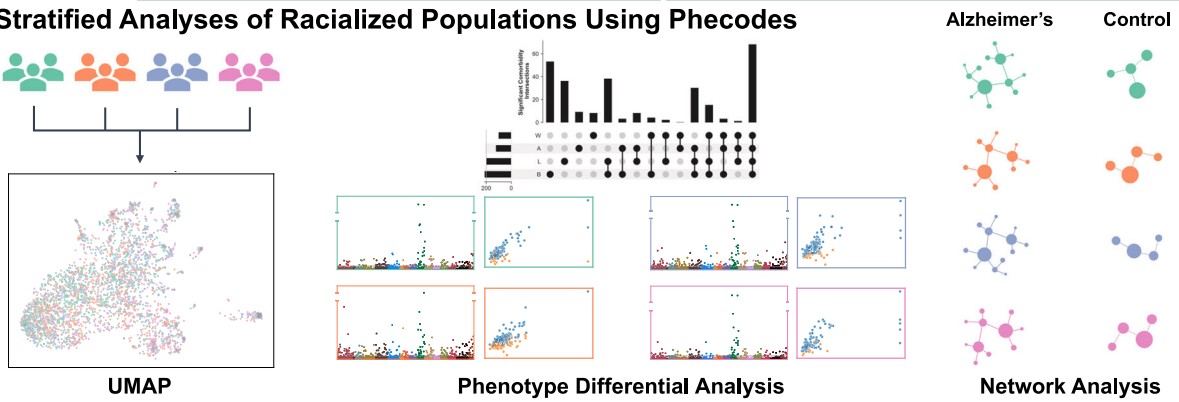

**Fig. 1 Visualization of analysis and methods.** This provides an overview of patient selection for UCSF and the UC-wide validation cohort, as well as the analyses performed, which include low-dimensional embedding, differential analysis, and network analysis. EMR Electronic medical records, OMOP Observational Medical Outcomes Partnership, PS Propensity score, R&E Race and ethnicity, UCD University of California Davis, UCI University of California Irvine, UCLA University of California Los Angeles, UCSD University of California San Diego, UCSF University of California San Francisco, UMAP Uniform Manifold Approximation and Projection. Green represents Asian-identified patients; orange represents Black-identified patients; purple represents Latine-identified patients, and pink represents White-identified patients.

**Table 1 Overview of patient demographics.**

| | UCSF | | | | UC-wide | | | |
| | Overall | AD | Control | SMD | Overall | AD | Control | SMD |
|---|---|---|---|---|---|---|---|---|
| n | 5064 | 1688 | 3376 | | 11,928 | 3976 | 7952 | |
| Sex, n (%) | | | | 0.018 | | | | 0.028 |
| Female | 3588 (70.9%) | 1196 (70.9%) | 2392 (70.9%) | | 8064 (67.6%) | 2684 (67.5%) | 5380 (67.7%) | |
| Male | 1476 (29.1%) | 492 (29.1%) | 984 (29.1%) | | 3864 (32.4%) | 1292 (32.5%) | 2572 (32.3%) | |
| Estimated age, mean (SD) | 87.3 (6.1) | 87.3 (6.1) | 87.3 (6.1) | 0.027 | 84.2 (6.0) | 84.3 (6.0) | 84.2 (6.0) | 0.020 |
| Identified R&E, n (%) | | | | | | | | |
| Asian | 1266 (25%) | 422 (25%) | 844 (25%) | | 2982 (25%) | 994 (25%) | 1988 (25%) | |
| Black | 1266 (25%) | 422 (25%) | 844 (25%) | | 2982 (25%) | 994 (25%) | 1988 (25%) | |
| Latine | 1266 (25%) | 422 (25%) | 844 (25%) | | 2982 (25%) | 994 (25%) | 1988 (25%) | |
| White | 1266 (25%) | 422 (25%) | 844 (25%) | | 2982 (25%) | 994 (25%) | 1988 (25%) | |
| Death status, n (%) | | | | 0.015 | | | | 0.054 |
| Alive | 4449 (87.9) | 1483 (87.9) | 2966 (87.9) | | 6814 (57.1) | 2267 (57.0) | 4547 (57.2) | |
| Deceased | 615 (12.1) | 205 (12.1) | 410 (12.1) | | 5114 (42.9) | 1709 (43.0) | 3405 (42.8) | |
| UC-wide Location, n (%) | | | | N.A. | | | | 0.016 |
| UCD | N.A. | N.A. | N.A. | | 2998 (25.1%) | 999 (25.1%) | 1999 (25.1%) | |
| UCI | N.A. | N.A. | N.A. | | 720 (6.0%) | 240 (6.0%) | 480 (6.0%) | |
| UCLA | N.A. | N.A. | N.A. | | 6497 (54.5%) | 2165 (54.5%) | 4332 (54.5%) | |
| UCSD | N.A. | N.A. | N.A. | | 1713 (14.4%) | 572 (14.4%) | 1141 (14.3%) | |

Summary of patients' estimated age, identified R&E, sex, and death status at UCSF and UC-wide, as well as location of care for UC-wide patients. *UCD* University of California Davis, *UCI* University of California Irvine, *UCLA* University of California Los Angeles, *UCSD* University of California San Diego, *UCSF* University of California San Francisco, *SD* Standard deviation, *N.A.* Not Applicable, *SMD* Standardized mean difference.

patients based on their overall phenotypic profiles, allowing us to assess whether there were global differences by AD status, R&E categories, and UC location. UMAP component means and standard deviations based on AD status, identified R&E, and UC location are provided in Supplementary Data 1.

First, a UMAP of patients' phenotypic profiles (UCSF: 1582 features, composed of phenotypes; UC-wide: 1521 features, composed of phenotypes) based on AD status was visualized (UCSF: Fig. 2a; UC-wide: Fig. 2g). Distributions of the first and second UMAP components were significantly different at UCSF

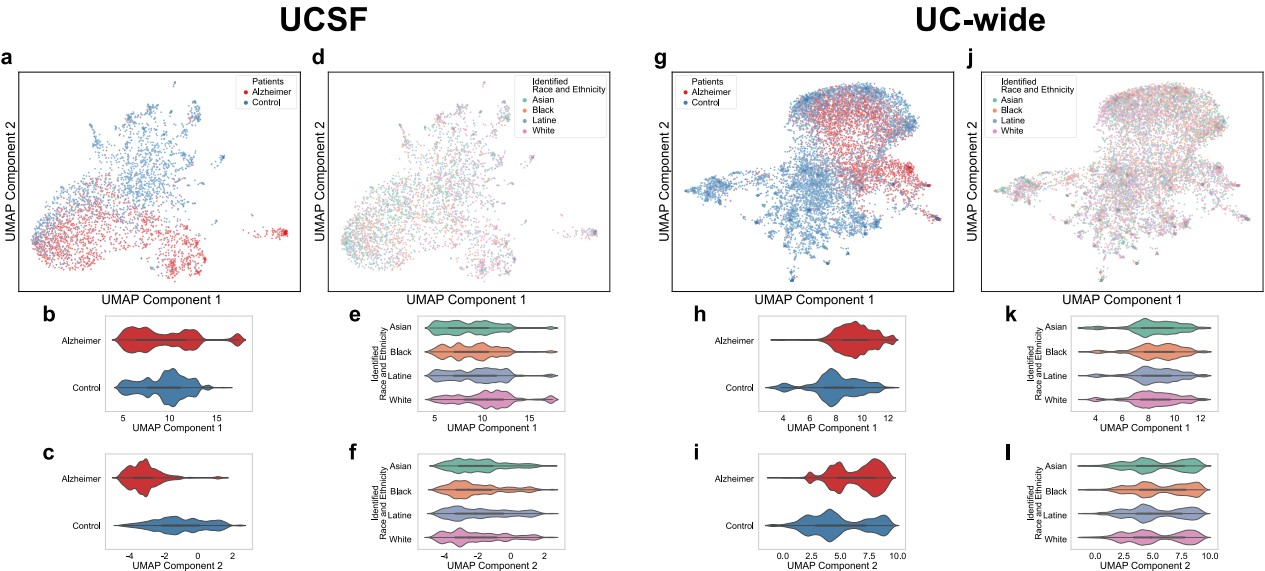

**Fig. 2 UMAPs visualizing patients' phenotypic profiles at UCSF and in the UC-wide validation cohort show separation based on AD status and identified R&E. a** UMAP of patients based on AD status at UCSF. **b** Distribution of the first UMAP component of patients based on AD status at UCSF. **c** Distribution of the second UMAP component of patients based on AD status at UCSF. **d** UMAP of patients based on identified R&E at UCSF. **e** Distribution of the first UMAP component of patients based on identified R&E at UCSF. **f** Distribution of the second UMAP component of patients based on identified R&E at UCSF. **g** UMAP of patients based on AD status UC-wide. **h** Distribution of the first UMAP component of patients based on AD status UC-wide. **i** Distribution of the second UMAP component of patients based on AD status UC-wide. **j** UMAP of patients based on identified R&E UC-wide. **k** Distribution of the first UMAP component of patients based on identified R&E UC-wide. **l** Distribution of the second UMAP component of patients based on identified R&E UC-wide. *P*-values to test differences in the distributions of UMAP components were computed using two-sided Mann-Whitney *U* tests for comparing patients based on AD status, while Kruskal-Wallis followed by post hoc two-sided Dunn's tests were used to compare patients based on identified R&E. For panels **a–f**, data were derived from *n* = 3737 patients at UCSF. For panels **g–l**, data were derived from *n* = 10,971 patients in the UC-wide validation cohort. For panels **a–c**, **g–i**, red = patients with AD; blue = control patients. For panels **d–f**, **j–l**, green = Asian-identified patients; orange = Black-identified patients; purple = Latine-identified patients; pink = White-identified patients. UMAP Uniform Manifold Approximation and Projection, UCSF University of California San Francisco, UC University of California, AD Alzheimer's dementia, R&E race and ethnicity, UMAP Uniform Manifold Approximation and Projection.

(two-sided Mann-Whitney *U* test - first component: statistic = 1.61e6, *p*-value = 7.7e-5; second component: statistic = 4.43e5, *p*-value = 0) (Fig. 2b, c) and UC-wide (two-sided Mann-Whitney *U* test - first component: statistic = 7.34e6, *p*-value = 0; second component: statistic = 8.81e6, *p*-value = 9.4e-225) (Fig. 2h, i) based on disease status.

Second, an analogous UMAP visualization was created of patients based on identified R&E (UCSF: Fig. 2d; UC-wide: Fig. 2j). Distributions of the first and second UMAP components were significantly different at UCSF (Fig. 2e, f) (Kruskal-Wallis test - first component: statistic = 171.00, *p*-value = 7.8e-37; second component: statistic = 14.23, *p*-value = 2.6e-3) and UC-wide (Fig. 2k, l) (Kruskal-Wallis test - first component: statistic = 40.77, *p*-value = 7.3e-9; second component: statistic = 45.00, *p*-value < 9.2e-10) between racialized populations. Validated significant differences using two-sided Dunn's tests were found for the first UMAP component between Black- and Latine-identified patients at UCSF (Fig. 2e) (statistic = −3.36; Bonferroni-corrected *p*-value = 2.3e-3) and in the UC-wide validation cohort (Fig. 2k) (statistic = 4.14; Bonferroni-corrected *p*-value = 1e-4). We also found significant differences using two-sided Dunn's tests for the first UMAP component between Black- and White- identified patients at UCSF (Fig. 2e) (statistic = −9.84; Bonferroni-corrected *p*-value = 0) and UC-wide (Fig. 2k) (statistic = 6.20; Bonferroni-corrected *p*-value = 0). This suggests that Black-identified patients might have, to an extent, distinct phenotypes relative to other racialized populations.

Finally, an analogous UMAP visualization was created of patients based on UC location for the UC-wide analysis

(Supplementary Fig. 3a). Distributions of the first (Supplementary Fig. 3b) and second (Supplementary Fig. 3c) UMAP components were significantly different (Kruskal-Wallis test - first component: statistic = 33.74, *p*-value = 2.3e-7; second component: statistic = 117.63, *p*-value = 2.5e-25). For the first UMAP component, significant differences were found between patients from UCLA and UCD (statistic = 3.37; Bonferroni-corrected *p*-value = 2.2e-3), UCLA and UCI (statistic = 3.56; Bonferroni-corrected *p*-value = 1.1 e-3), and UCLA and UCSD (statistic = −4.73; Bonferroni-corrected *p*-value = 0) using two-sided Dunn's tests. For the second UMAP component, significant differences were found between patients from UCD and UCI (statistic = 7.69; Bonferroni-corrected *p*-value = 0), UCD and UCLA (statistic = 7.18; Bonferroni-corrected *p*-value = 0), UCD and UCSD (statistic = 9.20; Bonferroni-corrected *p*-value = 0), UCLA and UCI (statistic = −4.07; Bonferroni-corrected *p*-value = 1e-4), and UCLA and UCSD (statistic = 4.34; Bonferroni-corrected *p*-value = 0) using two-sided Dunn's tests. Overall, UCLA patients' phenotypes significantly differ from the other three UC locations' patients' phenotypes for both UMAP components, while UCD patients' phenotypes significantly differ from the other three UC locations' patients' phenotypes for the second UMAP component. These findings support matching control patients to patients with AD by UC location in addition to the other covariates for the UC-wide analysis.

Based on the findings that there were global differences between patients' phenotypes based on identified R&E and AD status, we explored which phenotypes were significantly different between patients with AD and matched control patients for each racialized population.

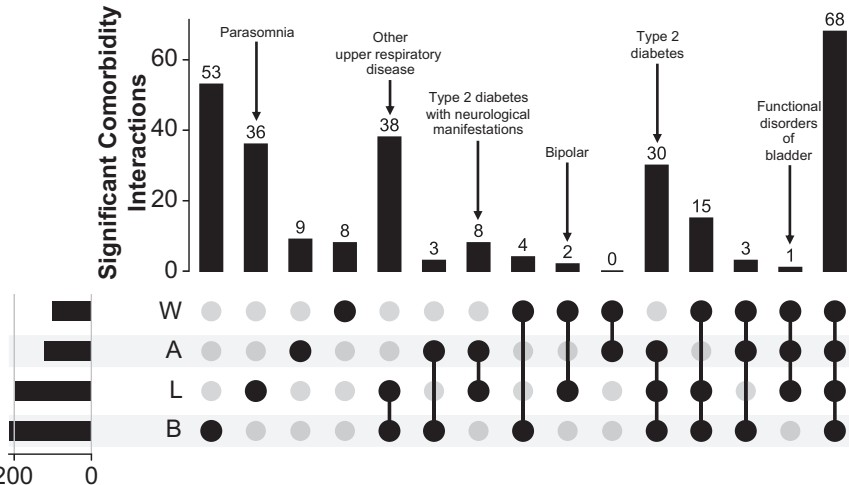

**Fig. 3 AD-associated comorbidities are often significant for all racialized populations.** Upset plot of significant comorbidity intersections across racialized populations with AD at UCSF. Rows indicate each racialized population analyzed. Bar chart shows single and overlapping significant comorbidities across racialized populations. AD Alzheimer's dementia, A Asian-identified patients, B Black-identified patients, L Latine-identified patients, W White-identified patients, AD Alzheimer's dementia, UCSF University of California San Francisco. Statistical significance was determined using two-sided Fisher's exact or chi-squared tests, comparing 931 phenotypes between patients with AD ($n = 422$) and control patients ($n = 844$) within each racialized population. Significance corresponds to a Bonferroni-corrected $p$-value < 0.05.

**Differential analysis shows that the majority of significant comorbidities shared by all racialized populations with AD at UCSF are validated in the UC-wide validation cohort.** Most comorbidities statistically significant for all racialized populations with AD relative to matched controls at UCSF are also significant in the validation cohort. For AD versus control differential analysis stratified by identified R&E, we found 68 shared significant comorbidities, represented as phenotypes, amongst all racialized populations with AD relative to controls at UCSF (Fig. 3, Supplementary Data 2). These include expected comorbidities such as neurological disorders, memory loss, vascular dementia, persistent mental disorders due to conditions classified elsewhere, and urinary tract infection (Fig. 4). Other shared significant comorbidities include major depressive disorder, anxiety disorder, psychosis, cerebrovascular disease, hypertension, and multiple osteoarthrosis comorbidities (two-sided Fisher's exact or chi-squared test, Bonferroni-corrected $p$-value < 0.05, Supplementary Data 2). Top categories include mental disorders ($n = 18$), symptoms phenotypes ($n = 7$), musculoskeletal phenotypes ($n = 7$), genitourinary phenotypes ($n = 5$), circulatory system phenotypes ($n = 5$), and endocrine/metabolic phenotypes ($n = 4$). Sixty-two out of 68 shared significant comorbidities (91.18%) amongst all racialized populations with AD at UCSF were also statistically significant in the UC-wide validation cohort for all racialized populations (Supplementary Data 3, Supplementary Fig. 4).

**When stratified by identified R&E, the odds ratios of comorbidities significant in patients with AD relative to matched controls at UCSF and in the UC-wide validation cohort are significantly correlated.** When stratifying significant AD-associated comorbidities by identified R&E, we found significant correlation between the odds ratios of these comorbidities at UCSF and UC-wide when significant comorbidities at UCSF are also significant UC-wide. For each racialized population, all

significant comorbidities found at UCSF were also found in the UC-wide validation cohort. First, for Asian-identified patients with AD, 102 of the 122 (83.61%) significant comorbidities at UCSF were also significant in the UC-wide cohort (two-sided Spearman's $\rho = 0.78$, $p$-value = 2.2e-22, Fig. 5a). Second, for Black-identified patients with AD, 138 of the 213 (64.79%) significant comorbidities at UCSF were also significant in the UC-wide cohort (two-sided Spearman's $\rho = 0.73$, $p$-value = 5.4e-24, Fig. 5b). Third, for Latine-identified patients with AD, 168 of the 198 (84.85%) significant comorbidities at UCSF were also significant in the UC-wide cohort (two-sided Spearman's $\rho = 0.69$, $p$-value = 1.8e-25, Fig. 5c). Finally, for White-identified patients with AD, 90 of the 100 (90%) significant comorbidities at UCSF were also significant in the UC-wide cohort (two-sided Spearman's $\rho = 0.68$, $p$-value = 1.4e-13, Fig. 5d). By contrast, for comorbidities significant only at UCSF, we found no significant correlation for most racialized populations, with the exception of Black-identified patients (two-sided Spearman's $\rho = 0.47$, $p$-value = 2.5e-5, Fig. 5c). Overall, these findings suggest concordance between the odds ratios of validated AD-associated comorbidities across the two EMR systems.

**Diabetes-related conditions, bipolar disorder, upper respiratory disease, and other comorbidities shared by a subset of racialized populations with AD at UCSF are validated in the UC-wide cohort.** After identifying significant comorbidities for each racialized population from stratified phenotype differential analysis, we also explored whether there were any overlapping significant comorbidities specific to one racialized population or shared by each combination of two or three racialized populations with AD at UCSF (Fig. 3). When identifying significant comorbidities specific to 1 racialized population with AD, we found 53 significant comorbidities specific to Black-identified patients, 36 specific to Latine-identified patients, 9 specific to Asian-identified patients, and 8 specific to White-identified

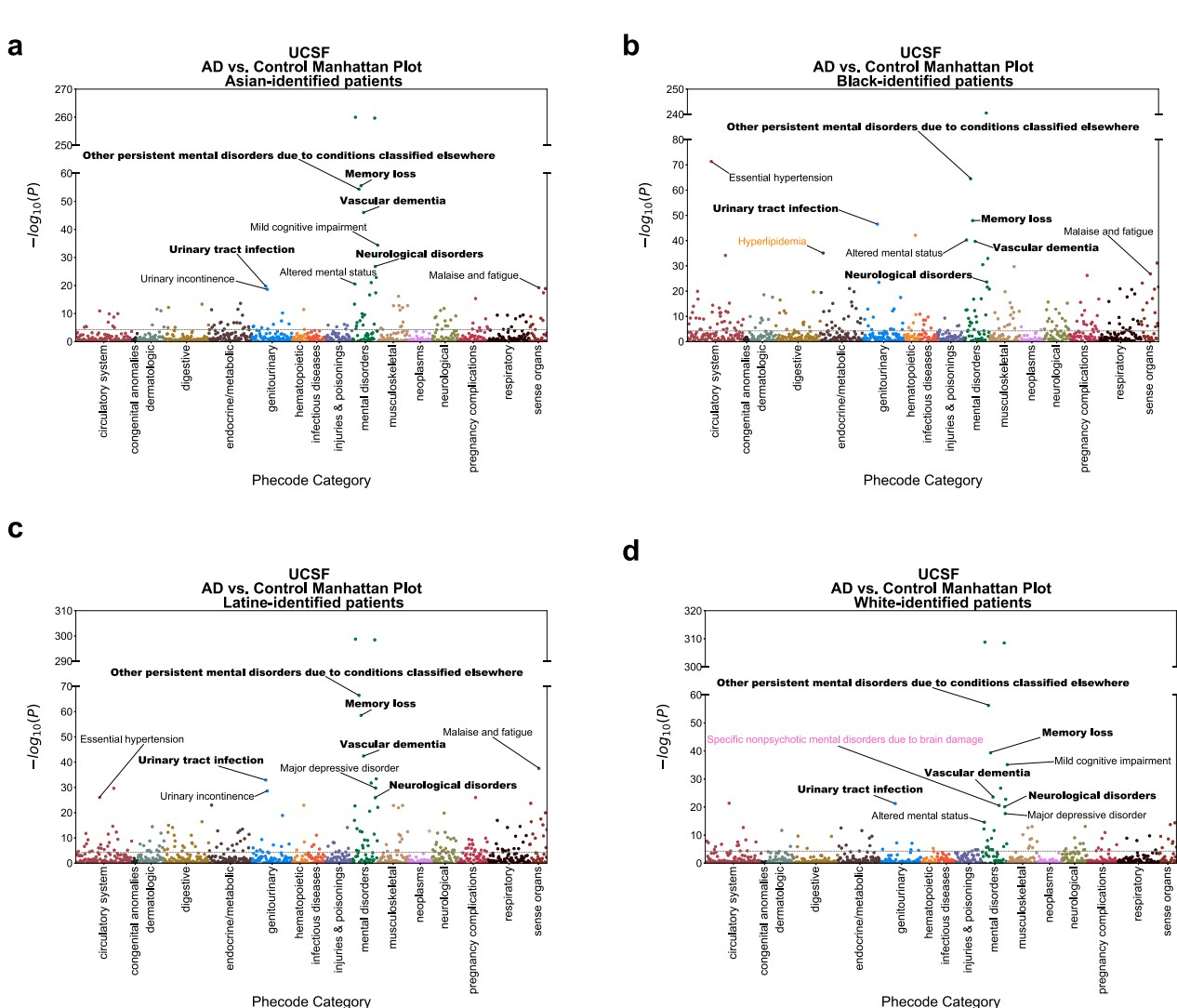

**Fig. 4 Differential analysis reveals that top significant comorbidities often overlap for multiple racialized populations with AD.** Manhattan plots of significant comorbidities, represented as phecode-corresponding phenotypes, colored by phecode categories, for (**a**) Asian-, (**b**) Black-, (**c**) Latine-, and (**d**) White- identified patients with AD. Statistical significance was determined using two-sided Fisher's exact or chi-squared tests, comparing 931 phenotypes between patients with AD ($n = 422$) and control patients ($n = 844$) within each racialized population at UCSF. Significance corresponds to a Bonferroni-corrected $p$-value < 0.05. Nine of the most significant comorbidities are annotated. Bold black annotation = top significant comorbidities shared by all racialized populations; black annotation = top significant comorbidities shared by multiple racialized populations; orange annotation = one of the top significant comorbidities for Black-identified patients; pink annotation = one of the top significant comorbidities for White-identified patients. AD Alzheimer's dementia, UCSF University of California San Francisco.

patients. When identifying significant comorbidities specific to 2 racialized populations with AD, we found that the majority of overlapping comorbidities were shared between Black- and Latine- identified patients, who shared 38 significant comorbidities. Additionally, we found 8 significant comorbidities specific to Asian- and Latine- identified patients, 4 specific to Black- and White- identified patients, 3 specific to Asian- and Black- identified patients, and 2 specific to Latine- and White- identified patients. When identifying significant comorbidities specific to 3 racialized populations with AD, we found that the majority of overlapping comorbidities were shared between Asian-, Black-, and Latine- identified patients, who shared 30 significant comorbidities. Additionally, we found 15 significant

comorbidities specific to Black-, Latine-, and White- identified patients, 3 specific to Asian-, Black-, and White- identified patients, and 1 specific to Asian-, Latine-, and White- identified patients.

Only a few comorbidities that were significant in subsets of racialized populations with AD at UCSF were validated UC-wide (Supplementary Fig. 4). For significant comorbidities specific to 1 racialized population with AD at UCSF, we found that 6 out of 36 (16.67%) significant comorbidities specific to Latine-identified patients are also significant in the validation cohort; these comorbidities are parasomnia, tinnitus, otalgia, hyperhidrosis, disturbance of skin sensation, and open wounds of head, neck, and trunk. For significant comorbidities specific to 2 racialized

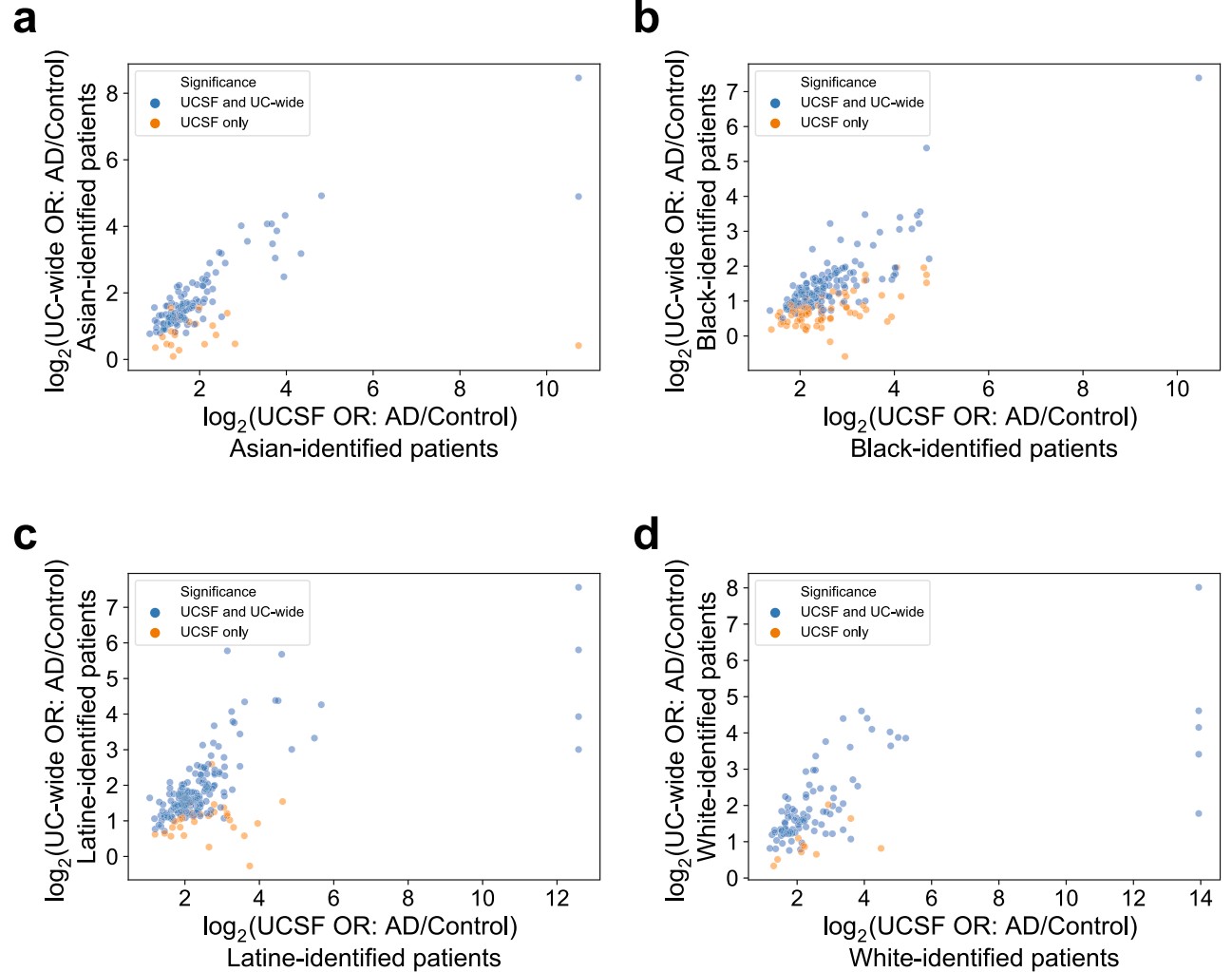

**Fig. 5 Significant AD-associated comorbidities at UCSF that are also significant in the UC-wide validation cohort are significantly correlated.** AD versus control log-log odds ratio correlation plots of phenotypes for (**a**) Asian-, (**b**) Black-, (**c**) Latine-, and (**d**) White- identified patients with AD between UCSF and UC-wide. Comorbidities (represented as phecode-corresponding phenotypes) shown are significant at UCSF (two-sided Fisher's exact or chi-squared tests, Bonferroni-corrected *p*-value < 0.05). Each dot represents a comorbidity. Dots in blue represent comorbidities significant both at UCSF and in the UC-wide validation cohort, while dots in orange represent comorbidities significant at UCSF only. AD Alzheimer's dementia, OR Odds ratio, UCSF University of California San Francisco. For Asian-identified patients (**a**), 102 phenotypes were significant at both UCSF and UC-wide, while 20 were significant only at UCSF. For Black-identified patients (**b**), 138 phenotypes were significant at both UCSF and UC-wide, while 75 were significant only at UCSF. For Latine-identified patients (**c**), 168 phenotypes were significant at both UCSF and UC-wide, while 30 were significant only at UCSF. For White-identified patients (**d**), 90 phenotypes were significant at both UCSF and UC-wide, while 10 were significant only at UCSF. For each racialized population, *n* = 422 patients with AD at UCSF; *n* = 844 control patients at UCSF; *n* = 994 patients with AD in the UC-wide validation cohort; *n* = 1988 control patients in the UC-wide validation cohort.

populations with AD at UCSF, we found that 2 out of 38 (5.26%) significant comorbidities specific to Black- and Latine-identified patients are also significant in the validation cohort; these comorbidities are upper respiratory disease and inflammation of eyelids. Additionally, type 2 diabetes with neurological manifestations is the only significant comorbidity out of 8 (12.5%) specific to Asian- and Latine- identified patients that is also significant in the validation cohort, while bipolar is the only significant comorbidity out of 2 (50%) specific to Latine- and White-identified patients that is also significant in the validation cohort. For significant comorbidities specific to 3 racialized populations with AD at UCSF, we found that acquired hypothyroidism is the only significant comorbidity out of 15 (6.67%) specific to Black-, Latine- and White- identified patients that is also significant in the validation cohort. Additionally, we found that functional disorders of bladder, which is the only significant comorbidity

specific to Asian-, Latine-, and White- identified patients at UCSF, is also significant in the validation cohort. Finally, type 2 diabetes is the only significant comorbidity out of 30 (3.33%) that we found specific to Asian-, Black-, and Latine- identified patients that is also significant in the validation cohort. Overall, these findings suggest that there are only a few significant comorbidities specific to racialized populations at UCSF that are also significant for these same specific populations in the UC-wide validation cohort.

**Network metrics reveal differences between AD networks, control networks, and AD vs control networks when stratified by patients' identified R&E.** We then generated identified R&E-stratified AD and control phenotype networks, all thresholded by 5% shared nodes (phenotypes) and 5% shared edges (phenotype pairs) for the UCSF cohort (Supplementary Data 4,

Supplementary Data 5). First, we analyzed and compared network metrics that quantify network topology and how nodes are connected and arranged. We explored whether network metrics are significantly different between identified R&E-stratified AD networks, identified R&E-stratified control networks, and identified R&E-stratified AD networks relative to networks from matched controls (Supplementary Data 6). At UCSF, all identified R&E-stratified AD phenotype networks have more nodes and edges compared to their respective control networks. When comparing network metrics between identified R&E-stratified AD networks using Kruskal-Wallis tests, the following metrics were found to be significantly different between them: average shortest path length (statistic = 19.33, $p$-value = 2.3e-4), closeness centrality (statistic = 19.33, $p$-value = 2.3e-4), degree (statistic = 77.76, $p$-value = 9.3e-17), neighborhood connectivity (statistic = 133.62, $p$-value = 9.0e-29), radiality (statistic = 425.67, $p$-value = 6.1e-92), and stress (statistic = 9.67, $p$-value = 2.2e-2), (Fig. 6a–d, Supplementary Fig. 5). The following network metrics were also found to be significantly different between identified R&E stratified control networks using Kruskal-Wallis tests: degree (statistic = 36.94, $p$-value = 4.7e-8), neighborhood connectivity (statistic = 125.82, $p$-value = 4.3e-27), radiality (statistic = 115.47, $p$-value = 7.3e-25), and stress (statistic = 9.74, $p$-value = 2.1e-2) (Supplementary Fig. 6). Additionally, we found that clustering coefficient (statistic = 20.41, $p$-value = 1.4e-4), eccentricity (statistic = 62.46, $p$-value = 1.7e-13), and topological coefficient (statistic = 20.25, $p$-value = 1.5e-4) were significantly different between these control networks at UCSF using Kruskal-Wallis tests (Supplementary Fig. 6). Finally, when comparing node-level network metrics between AD networks and matched control networks at UCSF using two-sided Mann-Whitney $U$ tests, we found that the majority of network metrics that were significantly different between AD and matched control networks were significantly different for all racialized populations. Clustering coefficient, degree, neighborhood connectivity, radiality, and topological coefficient were significantly higher for all four AD networks relative to respective control networks (Supplementary Data 7). We also found that average shortest path length, closeness centrality, and eccentricity were significantly different for networks representing Asian-, Latine-, and White- identified patients with AD relative to their respective control networks (Supplementary Data 7). Finally, we found that stress is significantly different for networks representing Black- and Latine-identified patients with AD relative to respective control networks (Supplementary Data 7).

Second, we asked whether top phenotype pairs shared between UCSF and the UC-wide validation cohort differed between identified R&E-stratified AD networks, as well as between identified R&E-stratified control networks. We found that 4 of the top 10 phenotype pairs were shared across all AD networks (Supplementary Data 8). These pairs include AD and dementias, essential hypertension, and hyperlipidemia (separately), as well as dementias and hypertension. Pain in joint and AD, and pain in joint and dementias, as well as dementias and malaise and fatigue, were found to be top pairs specific to Asian-identified patients; these phenotype pairs were shared by 40.5%, 40.3%, and 43.4% of patients at UCSF and by 38.9%, 38.3%, and 40.1% of patients in the UC-wide validation cohort, respectively. Meanwhile, other anemias and AD, as well as other anemias and dementias, were top pairs specific to Black-identified patients; these phenotype pairs were shared by 43.1% and 42.9% of patients at UCSF and by 40.7% and 40.3% of patients in the UC-wide validation cohort, respectively. Dementias and memory loss was a top pair for White-identified patients; this phenotype pair was shared by 25.8% of patients at UCSF and by 52.3% of patients in the UC-wide validation cohort. There were no top pairs unique to networks representing Latine-identified patients with AD.

Three of the top pairs in identified R&E-stratified control networks were shared across all networks (Supplementary Data 9). These pairs include essential hypertension and hyperlipidemia, malaise and fatigue, and pain in joint (separately). Essential hypertension and cough and osteoporosis (separately), as well as hyperlipidemia and type 2 diabetes, were found to be top pairs specific to Asian-identified control patients. Essential hypertension and acute renal failure was found to be a top pair specific to Black-identified control patients, while essential hypertension and gastroesophageal reflux disease (GERD) was a top pair unique to Latine-identified control patients. Finally, essential hypertension and atrial fibrillation was a top pair unique to White-identified control patients.

Third, we compared overall AD and control network metrics between UCSF and the UC-wide validation cohort. We found that AD and control network metrics at UCSF and UC-wide, standard normalized by the metric, were significantly correlated (two-sided Spearman's $\rho$ = 0.64, $p$-value = 2.1e − 10) (Fig. 6e, f). This suggests that network metrics are similar between corresponding AD and control networks across the two EMR systems.

Finally, we explored which network metrics were also significantly different between AD networks, control networks, and AD versus matched control networks in the UC-wide validation cohort (Supplementary Data 6, 10–11). Like UCSF, identified R&E-stratified AD networks in the validation cohort have more nodes and edges compared to their respective control networks when thresholded by 5% shared nodes and 5% shared edges. This suggests that AD networks are generally relatively more connected, with many shared phenotypes amongst patients within each network.

For network metrics that were significantly different between AD networks at UCSF, we found that only radiality is also significantly different in the validation cohort using Kruskal-Wallis tests (statistic = 88.87, $p$-value = 3.8e-19). Specifically, we found using two-sided Dunn's tests that AD networks representing Latine-identified patients have higher radiality relative to AD networks representing Black- identified patients at UCSF (statistic = −3.88; Bonferroni-corrected $p$-value = 3e-4) and in the UC-wide validation cohort (statistic = −5.41; Bonferroni-corrected $p$-value = 0). Similarly, we also found using two-sided Dunn's tests that Latine-identified patients have higher radiality relative to White- identified patients at UCSF (statistic = 14.90; Bonferroni-corrected $p$-value = 0) and UC-wide (statistic = 5.35; Bonferroni-corrected $p$-value = 0). This suggests that network metric differences found between AD networks at UCSF were, for the most part, not validated in the UC-wide cohort.

Using Kruskal-Wallis tests, we found that the following 3 network metrics that were significantly different between stratified control networks at UCSF were also significantly different in the UC-wide validation cohort: eccentricity (statistic = 265.27, $p$-value = 3.3e-57), radiality (statistic = 223.87, $p$-value = 2.9e-48), and topological coefficient (statistic = 13.74, $p$-value = 3.3e-3). When testing for differences in post hoc pairwise comparisons using two-sided Dunn's tests, we found that eccentricity was significantly higher for Asian- relative to Black- identified patients' control networks at UCSF (statistic = 4.41; Bonferroni-corrected $p$-value = 0) and in the UC-wide validation cohort (statistic = 12.49; Bonferroni-corrected $p$-value = 0). We also found that radiality is significantly higher for Asian- relative to Latine- identified patients' control networks at UCSF (statistic = 6.73; Bonferroni-corrected $p$-value = 0) and in the UC-wide validation cohort (statistic = 6.59; Bonferroni-corrected $p$-value = 0) using two-sided Dunn's tests. Radiality was also found to be significantly higher for Asian- relative to White- identified patients at UCSF (statistic = 5.55; Bonferroni-

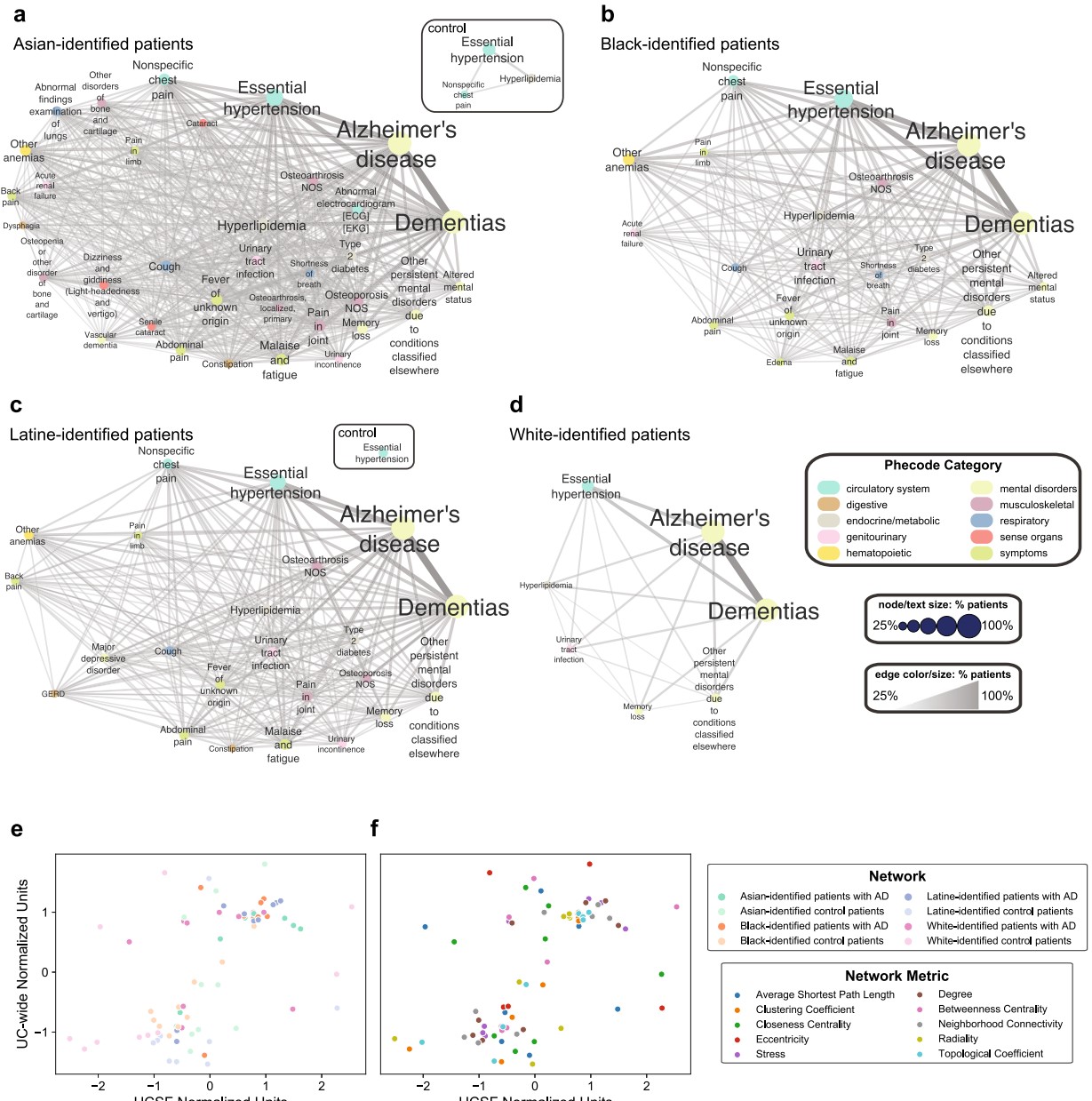

**Fig. 6 Phenotype network visualization of patients, stratified by identified R&E.** Phenotype networks for (**a**) Asian-, (**b**) Black-, (**c**) Latine-, and (**d**) White- identified patients with AD ($n = 422$ patients per identified R&E category) who received care at UCSF are thresholded by 25% of shared phenotypes for visualization. When applicable, phenotype networks of UCSF-based control patients stratified by identified R&E, thresholded by 25% of shared phenotypes for visualization, are also shown ($n = 844$ patients per identified R&E category). **e**, **f** Correlation plot of z-score normalized network metrics between UCSF and UC-wide, with colors representing network (**e**) or network metric (**f**) (two-sided Spearman's ρ = 0.64, p-value = 2.1e − 10). For panels **a–d**, phecode categories are colored as follows: teal = circulatory system; brown = digestive system; gray = endocrine/metabolic; pink = genitourinary; yellow = hematopoietic; light green = mental disorders; purple = musculoskeletal; blue = respiratory; red = sense organs; green = symptoms. Node and text size correspond to percentages of patients with a given phenotype ranging from 25 to 100%. Edge color and size correspond to percentages of patients sharing a phenotype pair, ranging from 25 to 100%. At UCSF, at least 5% of Asian-identified patients with AD shared 270 nodes, while at least 5% of Asian-identified control patients shared 111 nodes (**a**). At least 5% of Black-identified patients with AD shared 234 nodes, while at least 5% of Black-identified control patients shared 30 nodes (**b**). At least 5% of Latine-identified patients with AD shared 244 nodes, while at least 5% of Latine-identified control patients shared 42 nodes (**c**). At least 5% of White-identified patients with AD shared 142 nodes, while at least 5% of White-identified control patients shared 9 nodes (**d**). For panel **e**, green = Asian-identified patients with AD; light green = Asian-identified control patients; orange = Black-identified patients with AD; light orange = Black-identified control patients; purple = Latine-identified patients with AD; light purple = Latine-identified control patients; pink = White-identified patients with AD; light pink = White-identified control patients. For each of the 8 networks represented, 10 network metrics were compared between UCSF and UC-wide validation cohort. For panel **f**, blue = Average Shortest Path Length; orange = Clustering Coefficient; green = Closeness Centrality; red = Eccentricity; purple = Stress; brown = Degree; pink = Betweenness Centrality; gray = Neighborhood Connectivity; light green = Radiality; cyan = Topological Coefficient. For each of the 10 network metrics represented, 8 networks were compared between UCSF and UC-wide validation cohort. AD Alzheimer's dementia, UCSF University of California San Francisco.

corrected $p$-value = 0) and UC-wide (statistic = 2.90; Bonferroni-corrected $p$-value = 1.1e-2) using two-sided Dunn's tests. Finally, we found that topological coefficient was significantly higher for Asian- relative to White- identified patients' control networks using two-sided Dunn's tests at UCSF (statistic = 4.12; Bonferroni-corrected $p$-value = 1e-4) and UC-wide (statistic = 2.73; Bonferroni-corrected $p$-value = 1.9e-2).

We found that all 5 network metrics found to be significantly higher between all identified R&E-stratified AD networks relative to matched control networks at UCSF - clustering coefficient, degree, neighborhood connectivity, radiality, and topological coefficient - were also found to be significantly higher for all comparisons in the UC-wide validation cohort (two-sided Mann-Whitney $U$ test, Supplementary Data 7). Thus, network metrics found to be significantly different for all racialized populations with AD relative to respective controls were validated in the UC-wide analysis, similar to how AD-associated comorbidities shared between all racialized populations were largely also significant in the validation cohort. Using two-sided Mann-Whitney $U$ tests, we also found that closeness centrality was higher for Asian-identified AD relative to matched control patients' networks at UCSF (statistic = 1.69e4; $p$-value = 4.8e-2) and in the UC-wide validation cohort (statistic = 1.14e4; $p$-value = 8.9e-3), while average shortest path length was lower at UCSF (statistic = 1.31 e4; $p$-value = 4.8e-2) and in the UC-wide validation cohort (statistic = 7.69e3; $p$-value = 8.9e-3); eccentricity was also lower for Asian-identified AD relative to matched control patients' networks at UCSF (statistic = 8.2e3; $p$-value = 5.7e-32) and in the UC-wide validation cohort (statistic = 714; $p$-value = 9.3e-64). Finally, we found that stress is significantly higher between Latine-identified AD and their respective matched control patients' networks at UCSF (statistic = 6.18e3; $p$-value = 1.6e-2) and in the UC-wide validation cohort (statistic = 9.47e3; $p$-value = 2.5e-2). Overall, there were relatively few significantly different network properties specific to subsets of racialized populations with AD relative to controls that were validated UC-wide.

## Discussion

In this study, we utilized EMR to identify comorbidities associated with AD, stratified by four racialized populations. Additionally, we compared disease networks amongst patients with AD and controls for each racialized population. We also used phecode-corresponding phenotypes in this study to compare comorbidities between UCSF and the UC-wide validation cohort, demonstrating the utility of phecodes in validating findings between different EMR databases.

First, we asked whether patients' phenotypic profiles differed by AD status, identified R&E, and UC location using low-dimensional embedding (UMAP). For AD status, significant differences were found between patients with AD and controls along both UMAP components at UCSF and in the validation cohort, consistent with Tang et al.[7] This suggests that representing patients using ICD diagnoses mapped to phenotypes yield similar results to using ICD diagnoses directly, providing support for the use of phenotypes to investigate patient comorbidities. Next, while we found numerous differences between patients' phenotypic profiles based on identified R&E at UCSF, many of these differences were not corroborated in the validation cohort, with the exception that Black-identified patients' phenotypic profiles were found to be significantly different from Latine- and White- identified patients' phenotypic profiles along the first UMAP component. Whether the finding that Black-identified patients may have distinct phenotypes is generalizable, or why this would be the case, is unknown; however, this could be due to

the unique impact of anti-Black racism, such as persistent residential segregation, on Black-identified individuals[2,21,22]. Third, for the UC-wide validation cohort, we asked whether patients' phenotypic profiles differed based on location. Since we found significant differences between patients' phenotypic profiles at UCLA and the other UC locations for both UMAP components, as well as significant differences between UCD and the other UCs for the second UMAP component, we decided to also match control patients based on UC location for the validation cohort to minimize confounding bias.

For phenotype differential analysis, we explored whether there were comorbidities significant for patients with AD relative to matched controls, stratified by identified R&E. We found that most shared comorbidities that were significant for all identified R&E with AD relative to matched controls were significant at UCSF and in the validation cohort. Many of these comorbidities have been identified in previous studies as associated with[7] and/or a risk factor for AD or dementia generally. These include cardiovascular risk factors like hyperlipidemia and essential hypertension[23], psychiatric conditions such as depression[24], anxiety[25], and psychosis[26], genitourinary conditions such as urinary tract infection, and musculoskeletal conditions such as osteoporosis[27]. By contrast, few comorbidities significant in specific racialized populations with AD at UCSF were validated in the UC-wide cohort, which could be due to differences in how diagnoses are mapped from the source EMR to databases at UCSF and in UCDDP. Additionally, while UCSF's EMR contains patient information since 1982, UCDDP's EMR contains patient information since 2012, which may also contribute to differences in findings. Interestingly, in our study, type 2 diabetes was found to be significant specifically in Asian-, Black- and Latine- identified patients with AD at UCSF and in the validation cohort. Additionally, type 2 diabetes with neurological manifestations was found to be significant only for Asian- and Latine- identified patients at UCSF and in the validation cohort. Prior studies suggest that diabetes is generally an AD risk factor[28,29]. Also, while not statistically significant, we did find a higher percentage of White-identified patients with AD with type 2 diabetes relative to matched controls at UCSF and in the validation cohort, a finding consistent with the other racialized populations in our study as well as other studies investigating the relationship between type 2 diabetes and AD. That type 2 diabetes was not significantly more prevalent in White-identified patients with AD in our study may be due to having a relatively small sample size.

Interestingly, we found bipolar disorder to be significant specifically in Latine- and White- identified patients with AD. A meta-analysis study suggests that this psychiatric disorder may increase dementia risk[30,31]. However, bipolar disorder is often misdiagnosed, particularly for individuals with African ancestry who are often categorized as Black[32]. Additionally, studies suggest that Asian-identified individuals may underutilize mental health services, leading to potential underdiagnosis of mental disorders for these individuals[33,34]. Follow-up analyses that consider the possibility of misdiagnosis, both in general and in specific racialized populations, are needed to contextualize this finding.

Acquired hypothyroidism was found to be significant in Black-, Latine-, and White- identified patients with AD, which has been previously identified as an AD risk factor in women[35]. Additionally, inflammation of eyelids and other upper respiratory disease were significant for Black- and Latine-identified patients with AD. While, to our knowledge, inflammation of eyelids has not been previously found to be associated with AD or dementia generally, previous work has shown that there are socioeconomic disparities in lung health, and since socioeconomic status and racial categories are strongly associated due to the effects of racism, it is possible that the upper respiratory disease association

for Black- and Latine- identified patients with AD could be a reflection of socioeconomic disparities that disproportionately impact these individuals[2,36]. Evidence suggests a potential link between air pollution and dementia risk; follow up analyses can explore whether exposure to air pollution is associated with increased respiratory diagnoses and AD risk for these individuals[37].

We found six dermatological and sensory-associated significant comorbidities, such as parasomnia and tinnitus, that were specific to Latine-identified patients with AD at UCSF and in the UC-wide validation cohort. To our knowledge, these six comorbidities have been largely unexplored in the context of AD or dementia generally. A prior study suggested that tinnitus may be associated with increased AD incidence, and a review suggested a possible relationship with a form of parasomnia and dementia; overall, however, the potential relationships between these comorbidities and AD are generally poorly understood or unknown[38,39]. One possibility is that, even with Bonferroni correction, these significant comorbidities could be false positives. It is also possible that these significant comorbidities could reflect cultural differences in how symptoms are expressed, though caution must be taken with this interpretation due to high intraracial heterogeneity[40,41]. Differences in symptom expression have been explored in numerous studies in the context of neuropsychiatric conditions such as depression, anxiety, and obsessive-compulsive disorder, where symptom presentations often differ based on an individual's cultural and geographic context[42–45].

In general, we found that AD network metrics, stratified by identified R&E, are similar overall. Interestingly, we also found a few specific top phenotype pairs in common between UCSF and UC-wide AD networks stratified by identified R&E, such as pain in joint and AD and dementia (separately) for Asian-identified patients, as well as anemia and AD and dementia (separately) specific to Black-identified patients. This underscores that there are a few specific network-level differences in patients' comorbidities between racialized populations.

For control networks, we found that eccentricity, radiality, and topological coefficient significantly differed between identified R&E stratified control networks at UCSF and in the UC-wide validation cohort, and each racialized population had at least one top comorbidity pair specific to that population. Generally, that control networks differ is consistent with findings from Glicksberg et al., which demonstrated differences in disease networks representing Black-, Latine-, and White- identified patients' diagnoses[8].

When we compared AD phenotype networks with controls stratified by identified R&E, we found that network metrics measuring centrality were significantly different between AD and control networks. At both UCSF and in the validation cohort, clustering coefficient, degree, neighborhood connectivity, radiality, and topological coefficient were all found to be higher for AD relative to control networks. Degree and neighborhood connectivity were also found to be higher for patients with AD relative to controls in Tang et al.[7] That AD networks are relatively more connected than control networks is consistent with the expectation that cohorts defined by a specific disease would have more shared phenotypes relative to matched controls. In our study, for AD networks representing Asian-identified patients specifically, closeness centrality was higher, which is consistent with findings from Tang et al. for AD networks relative to control networks generally. This suggests that phenotypes in Asian-identified patients with AD are more closely connected to one another relative to controls. Finally, for AD networks representing Latine-identified patients at UCSF and in the UC-wide validation cohort, stress was found to be higher relative to matched control networks, consistent with findings from Tang et al.

between AD and control networks generally. This suggests that phenotypes are relatively more connected to one another for Latine-identified patients with AD relative to controls. In general, network metrics found to be significantly different for all patients with AD relative to controls at UCSF were also significantly different for the UC-wide validation cohort.

Our study has several limitations. We performed an association analysis, which is by design not testing for causality. Health equity researchers strongly emphasize the need for causal-based studies due to its concrete potential to identify interventions to mitigate health disparities[41,46]. Our study may serve as a hypothesis generation tool that can be leveraged by researchers to gain actionable insight into comorbidities identified in specific racialized populations and clarify these comorbidities' relationship with AD. Also, indicators of SDoH that likely contributed to differences we do see between racialized populations, such as insurance and Medicaid status, as well as how patients' SDoH compare to those living in the surrounding area in which they received care, were not analyzed. This is due to the current unavailability of SDoH indicators in the de-identified OMOP databases we used for this study. SDoH, which in the U.S. often differ between racialized populations because of exposures to racism, must be included in analyses to properly contextualize differences in health outcomes[41,46]. Furthermore, in the context of dementia, a previous study suggested that accounting for socioeconomic factors such as income and formal education level may mitigate differences in dementia incidence between Black- and White- identified individuals[47]. There are a number of challenges obtaining individual-level SDoH in the EMR[48], including the lack of systematic inclusion. However, several community-level SDoH can be assessed for patients with known addresses, including median income, food access[49], and environmental pollution[50], though using these broader SDoH indicators to describe individuals have important limitations[51]. We plan to identify strategies to properly incorporate SDoH in future work. There may also be differences in AD diagnosis prevalence, as well as when AD is diagnosed, between individuals based on identified R&E, which has indeed been suggested between Black- and White- identified individuals with AD[52]. We attempted to account for this by comparing diagnoses between patients with AD to matched controls, stratified by identified R&E, but there are caveats to consider to this approach as well, such as the lack of direct measures of exposures to racism, which likely differ between individuals within each identified R&E[41]. Furthermore, there are differences in the prevalence of other diseases as well between racialized populations, which we also attempted to account for by performing a stratified analysis for each identified R&E[4]. Another limitation is that we grouped patients by R&E categories, even though high intraracial heterogeneity is likely. Several researchers have argued that grouping together Latine-identified individuals, for instance, is not advisable because of substantial heterogeneity in ancestries and countries of origin[41,53]. This likely extends to other racialized populations as well; Black- and Asian- identified individuals are also highly heterogeneous in the context of countries of origin and formal education level[54–57]. In the context of dementia, a study found substantial heterogeneity of dementia comorbidities among Asian-identified patients when stratified by self-identified Asian subgroup[13]. Related to intraracial heterogeneity, we also did not study how additional identities - such as sex, gender, LGBTQIA + identity, and disability status - impact health outcomes within racialized populations[41]. Also, Indigenous patients, such as those identified as American Indian and Alaska Native, or as Native Hawaiian and Pacific Islanders, were not included due to lower numbers in the EMR. While some studies have included these patients to determine these populations' rate of dementia

incidence and survival after diagnosis, methods robust to smaller sample size will likely need to be routinely leveraged in order to systematically include these individuals[13,41,58]. Since it is essential to include as many individuals from as many populations as possible, we plan to address this limitation by utilizing these more robust methods in future work. Additionally, beyond intraracial heterogeneity, over 10% of people identify as multiracial in the U.S., suggesting that stratifying individuals by R&E categories may result in the exclusion of a substantial number of people who identify with more than one category[59]. We also included death status for PS matching to account for potential differences in death rates; however, matching on death status, which could have been a consequence of AD, could also have resulted in less healthy control patients. Furthermore, even though we matched on death status, we cannot know for certain whether controls do not have AD, due to the fundamental inaccessibility to the ground truth of what conditions patients actually have. Additionally, we did not control for the number of visits nor the length of time patients have been in the EMR, which could have an effect on the number of diagnoses a patient has and thus is a limitation to consider when interpreting findings. Finally, while we used Bonferroni-corrected p-value thresholds, it is nonetheless possible that some significant comorbidities may be false positives. This could also be one reason why some significant comorbidities in the UCSF cohort were not found to be significant in the UC-wide validation cohort.

Overall, our findings suggest that comorbidities and network metrics found to be significantly different for all racialized populations with AD relative to matched controls when stratified by identified R&E at UCSF are also significantly different in the UC-wide validation cohort. It is encouraging that many of these comorbidities are thought to be modifiable risk factors, suggesting that interventions may be applicable to all who are at risk of developing AD[60]. Future work would be required to identify how all individuals can have equitable access to these interventions, and/or whether interventions tailored to specific racially marginalized communities can help mitigate AD disparities.

## Data availability

The data that support the findings of this study are not openly available to individuals not affiliated with UCSF due to the sensitivity of the data, with the exception of collaborators. Individuals not affiliated with UCSF may set up an official collaboration with a UCSF-affiliated investigator by contacting the principal investigator, Marina Sirota (marina.sirota@ucsf.edu). Requests should be processed within a couple of weeks. UCSF-affiliated individuals may contact UCSF's Clinical and Translational Science Institute (ctsi@ucsf.edu) or the UCSF's Information Commons team for more information (Info.Commons@ucsf.edu) to access the UCSF EMR database. UCDDP is only available to UC researchers who have completed analyses in their respective UC first and have provided justification for scaling their analyses across UC health centers. Censored source data for phenotype differential analysis and network analysis used to create Figs. 3, 4, 6a–d, Supplementary Fig. 5, and Supplementary Fig. 6 are provided in Supplementary Data 2–5, 10, and 11.

## Code availability

Corresponding code for the study can be found at https://doi.org/10.5281/zenodo.7764948[61].

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

## Acknowledgements

Primary support was provided by grant numbers NIA R01AG060393 and R01AG057683 (S.W., A.T., T.O., M.S.). Additional support was provided by the Medical Scientist Training Program T32GM007618 (A.T.). The authors acknowledge the use of resources developed and supported by the UCSF IT Academic Research Systems and the UCSF Bakar Computational Health Sciences Institute Information Commons teams, and thank members of these teams for technical support. The authors also thank the Center for Data-driven Insights and Innovation at UC Health (CDI2; https://www.ucop.edu/uc-health/functions/center-for-data-driven-insights-and-innovations-cdi2.html), for its analytical and technical support related to use of the UC Health Data Warehouse. Additionally, we thank Maria Glymour and Jackie Roger for reviewing and providing suggestions for the manuscript. We would also like to thank Jimmy Phuong and all members of the Sirota lab for their input and advice.

## Author contributions

S.W., A.T., and M.S. designed the study, experiments, and analytic plan. S.W., A.T., T.O., K.Y., and M.S. interpreted results. S.W. carried out the analysis and wrote the manuscript, which was edited and reviewed by all the authors.

## Competing interests

The authors declare no competing interests.
