## [Peer Review File · Communications Medicine]

Reviewers' comments:

Reviewer #1 (Remarks to the Author):

This analysis uses a big data approach to examine differences in co-morbidities for AD by race/ethnicity using EHR from the UCSF OMOP database and a UC-wide validation cohort. It is a follow-up study to a previous paper by these authors establishing using these techniques to do deep phenotyping of AD using EHR (that paper examined sex differences). Comments for authors:

1. There are so many comparisons presented in the results that it is difficult for the reader to keep track of or get a clear picture of the overall story. That being said, the Discussion does well to summarize and contextualize the many findings. Perhaps some of this summarizing could be done in the results.
2. Certain results for enriched comorbidities by R&E group seem to indicate false positives based on multiple comparisons, such as "open wounds to head, neck, and torso" in Latine-identified patients... it's hard to think of a scenario where that would be a real phenotype of AD that is specific to Latine persons. This is a potential limitation of the hypothesis-free big data approach; though Bonferroni corrections were done to address multiple comparisons, these findings that pop up lacking biological lack face validity are difficult to explain. This should at least be addressed in the Limitation (even though it is made clear that causality is not established and this approach is hypothesis-generating).
3. Related to the above point, the authors do not discuss the possibility that the findings in the UCSF cohort that are not replicated in the validation cohort may be due to them being false positives due to multiple comparisons.
4. Are the findings showing that network metrics measuring centrality are higher in AD than controls particularly informative? By definition, this is a group defined by having a specific medical condition (AD) so wouldn't they be expected to have more shared conditions than a control group not based on any single health condition? Or perhaps this is as expected and simply confirmatory (in which case, it may help the reader out to say so explicitly).
5. While the purpose of the UMAP analysis for dimensionality reduction is clear, some discussion of what the 2 UMAP principle components represent phenotypically is warranted; it is hard for the reader to comprehend how the entire universe of comorbidities could be reduced to 2 components or what they could mean without guidance from the authors.
6. Figure 1 legend does not indicate which colors represent which R&E group.

Reviewer #2 (Remarks to the Author):

This paper uses network analysis to compare comorbidity between racialized populations with Alzheimer's disease. Using EHR data, the authors conducted network analysis of disease networks for these patient populations, which appears to be a first.

Merits:

1. Exploring differences in comorbidities by race is a significant contribution. While the paper has merits, there are some issues to consider.

2. Figure 3 and the network from figure 6 are constructive in displaying differences and overlaps between the disease networks.
3. Robust methods were included to compare networks and validate the phenotypes.
4. The level of detail provided (with a few additions discussed below) makes this work reasonably straightforward to reproduce.

Issues:

1. Text in figures are difficult to read.
2. From where are the racial categories coming? There is no indication of how Native Hawaiians/Pacific Islanders were classified. Also, why use the term Latine? That is non-standard and should be addressed.
3. There are a lot of network measures. However, a comprehensive discussion of the "so what" of these. Figure 6 includes many metrics, but how useful they will be is not indicated. Given that the number of nodes varies, it is evident that other network metrics will vary. The discussion seems to focus on comorbidities and differences between statistically significant ones, for which network analysis is unnecessary. Why not provide the prevalence for these comorbidities and combinations instead of the networks? Why is it important to note that Latine-identified patients with AD have more interconnected, centralized comorbidities? Repeating the results in the discussion does not add much.
4. It is well established that there are vast differences in utilization patterns by race, as racialized minorities are less likely to seek regular care and will have less comprehensive EHR data. This can introduce bias in EHR-based research, which must be addressed. Additionally, what is the timeframe in which the authors identified comorbidities? If some patients have been seen for longer in the healthcare system (those who seek care and/or are sicker), they will have more data than those that do not, and that can introduce bias. This becomes even more of an issue when you select controls. How can you be sure the controls really do not have Alzheimer's?
5. The researchers need to indicate the representativeness by race and gender of the dataset they derived their subsets from. They created equal-sized cohorts, but it is not clear how representative the underlying population is.

Reviewer #3 (Remarks to the Author):

This paper utilizes EMRs from the patients receiving care at UCSF to assess whether comorbidities differed by race (Asian, Non-Hispanic Black, Latino, Non-Hispanic white). The analyses were then replicated using patients receiving care at the larger University of California system (UC Davis, UC Irvine, UCLA, and UCSD). The authors report that although AD-associated comorbidities are generally similar across racialized populations, there are some differences. The paper is interesting and a potential starting point. However, there are several aspects that are not addressed, and the overall significance is difficult to interpret.

1. Please provide an overview of which patients are seen at each of the UC sites. Are the patients seen at these sites representative of the surrounding population? Are they of higher SES and therefore not as generalizable? Are patients on Medicaid seen? All of these factors are important in assessing racial/ethnic differences due to differences in access to care.

2. African Americans are diagnosed at later stages of dementia than whites (Lennon et al. 2021). Thus, the diagnosis depends on race. This makes it difficult to then compare comorbidities by race. Can the authors better respond to this?
3. Similarly, there are well-known racial/ethnic differences in the diagnoses of other conditions – how might this play a role in the results?
4. Can any information be obtained on education, income, or other social determinants of health?
5. In the discussion, the authors briefly mention that different cultures can present symptoms differently. This is an extremely important point and deserves more attention.
6. I am a bit confused on the matching by death status. Why not just include individuals living at the time – which I think was between Sept and Oct 2021?
7. Codes were included before and after the AD diagnoses – how many years before were codes included?
8. Was there a race/ethnic differences in number of medical visits? Presumably the more visits the more diagnoses.

Reviewer Comments

Referee #1: PhD, epidemiology, Alzheimers, social/lifestyle factors

Referee #2: PhD, data management, data mining, network analysis, co-morbidities, EHRs

Referee #3: PhD, epidemiology, neurodegenerative disease, progression markers

Reviewers' comments:

Reviewer #1 (Remarks to the Author):

This analysis uses a big data approach to examine differences in co-morbidities for AD by race/ethnicity using EHR from the UCSF OMOP database and a UC-wide validation cohort. It is a follow-up study to a previous paper by these authors establishing using these techniques to do deep phenotyping of AD using EHR (that paper examined sex differences).

We would like to thank the reviewer for summarizing our work.

Comments for authors:

1. There are so many comparisons presented in the results that it is difficult for the reader to keep track of or get a clear picture of the overall story. That being said, the Discussion does well to summarize and contextualize the many findings. Perhaps some of this summarizing could be done in the results.

We would like to thank the reviewer for the feedback and suggestion. We have restructured the results section to include summaries in order to better organize our findings and maintain a clearer picture of the overall story.

2. Certain results for enriched comorbidities by R&E group seem to indicate false positives based on multiple comparisons, such as “open wounds to head, neck, and torso” in Latine-identified patients... it’s hard to think of a scenario where that would be a real phenotype of AD that is specific to Latine persons. This is a potential limitation of the hypothesis-free big data approach; though Bonferroni corrections were done to address multiple comparisons, these findings that pop up lacking biological lack face validity are difficult to explain. This should at least be addressed in the Limitation (even though it is made clear that causality is not established and this approach is hypothesis-generating).

Thank you for bringing up this important issue of the possibility that some of the enriched comorbidities that we see could be due to false positives. To address this, we added the following to the discussion section on page 23: “One possibility is that, even with Bonferroni correction, these significant comorbidities could be false positives” as well as on page 26 in the limitations section of the discussion: “Finally, while we used Bonferroni-corrected p-value

thresholds, it is nonetheless possible that some of these significant comorbidities may be false positives.”

3. Related to the above point, the authors do not discuss the possibility that the findings in the UCSF cohort that are not replicated in the validation cohort may be due to them being false positives due to multiple comparisons.

Thank you for bringing up this possibility. To address and acknowledge this, we added the following to the limitations section of the discussion on page 26: “Finally, while we used Bonferroni-corrected p-value thresholds, it is nonetheless possible that some of these significant comorbidities may be false positives. This could also be one reason why some significant comorbidities in the UCSF cohort were not found to be significant in the UC-wide validation cohort.”

4. Are the findings showing that network metrics measuring centrality are higher in AD than controls particularly informative? By definition, this is a group defined by having a specific medical condition (AD) so wouldn't they be expected to have more shared conditions than a control group not based on any single health condition? Or perhaps this is as expected and simply confirmatory (in which case, it may help the reader out to say so explicitly).

We agree with the reviewer that it is expected that the AD cohort would be more homogeneous relative to matched controls. To address this, we added to the discussion on page 24: “That AD networks are relatively more connected than control networks is consistent with the expectation that cohorts defined by a specific disease would have more shared phenotypes relative to matched controls.”

5. While the purpose of the UMAP analysis for dimensionality reduction is clear, some discussion of what the 2 UMAP principle components represent phenotypically is warranted; it is hard for the reader to comprehend how the entire universe of comorbidities could be reduced to 2 components or what they could mean without guidance from the authors.

Thank you for the question. Yes, the two UMAP components represent the constellation of phenotypes that patients have in a low-dimensional space while allowing for the preservation of local and global distances. This enables us to visualize the patients based on their overall clinical profile, showing global differences by AD status, demographic factors, and location of care. Further unsupervised clustering approaches paired with association analysis would allow for the exploration of subgroups of patients and their associated clinical features; however, this is outside the scope of the current work. Since we wanted to compare all comorbidities between all patients with AD and controls within each racialized population, we did not take this targeted approach. We clarified the purpose of the UMAP analysis on page 10: “We performed low-dimensional embedding of patients using non-AD ICD codes aggregated into phecode-corresponding phenotypes (referred to as phenotypes hereafter) to visualize patients based on their overall phenotypic profile, allowing us to assess whether there were global differences by AD status, R&E categories, and UC location.”

6. Figure 1 legend does not indicate which colors represent which R&E group.

Thank you, this has been corrected by adding the following to the figure 1 legend: “Green represents Asian-identified patients; orange represents Black-identified patients; purple represents Latine-identified patients, and pink represents White-identified patients.”

Reviewer #2 (Remarks to the Author):

This paper uses network analysis to compare comorbidity between racialized populations with Alzheimer's disease. Using EHR data, the authors conducted network analysis of disease networks for these patient populations, which appears to be a first.

Merits:

1. Exploring differences in comorbidities by race is a significant contribution. While the paper has merits, there are some issues to consider.
2. Figure 3 and the network from figure 6 are constructive in displaying differences and overlaps between the disease networks.
3. Robust methods were included to compare networks and validate the phenotypes.
4. The level of detail provided (with a few additions discussed below) makes this work reasonably straightforward to reproduce.

We would like to thank the reviewer for summarizing our work and recognizing its merits.

Issues:

1. Text in figures are difficult to read.

Thank you, we did our best to increase the font and make the figure text as readable as possible. We also transferred Figures 6e and 6f to Supplementary Figures 5 and 6 to make these figures larger and more readable.

2. From where are the racial categories coming? There is no indication of how Native Hawaiians/Pacific Islanders were classified. Also, why use the term Latine? That is non-standard and should be addressed.

Thank you for raising the important question of where these racial categories come from. At UCSF, patients' identified race and ethnicity are primarily derived from self-identification based on an algorithm that is now cited (UCSF Health's equity-related variables user's guide) in the methods section under Patient Cohort Selection on page 6: “A UCSF algorithm that uses self-

identified R&E as inputs was primarily used to determine identified R&E (UCSF Health's equity-related variables user's guide. (2021))." Patients who self-identified or were identified as Native Hawaiians/Pacific Islanders were classified in a separate category as Native Hawaiian or Other Pacific Islander. These individuals were not included in the study due to small numbers.

For the UC-wide validation cohort, it is less clear where the racial categories are coming from, as this information is not provided. Therefore, it is possible that these categories come from self-identification or provider identification. A modified version of the UCSF algorithm that was referenced is described in the methods section on page 6: "A modified version of the UCSF algorithm for R&E categorization was used to identify R&E for the small number of patients that had an identified race and an identified ethnicity, but did not have a combined identified R&E. This version uses descriptions of patients' identified race and identified ethnicity in the database as inputs to determine patients' combined identified R&E."

The term Latine has been used recently in numerous publications to maintain gender inclusivity, which the term Latinx offers, while being more intuitive to pronounce in Spanish and other Romance languages. This has been clarified in the introduction on page 3: "we use Latine here as a gender-neutral term that is more intuitive to pronounce in Spanish and other Romance languages"

3. There are a lot of network measures. However, a comprehensive discussion of the "so what" of these. Figure 6 includes many metrics, but how useful they will be is not indicated. Given that the number of nodes varies, it is evident that other network metrics will vary. The discussion seems to focus on comorbidities and differences between statistically significant ones, for which network analysis is unnecessary. Why not provide the prevalence for these comorbidities and combinations instead of the networks? Why is it important to note that Latine-identified patients with AD have more interconnected, centralized comorbidities? Repeating the results in the discussion does not add much.

We thank the reviewer for these questions and suggestion. We think that including both the prevalence information as well as the network analysis is informative in different ways, so we chose to keep both. We think that the network analyses provide us with higher order representations of phenotypes that patients have that can then be visualized, measured, and compared.

We think that these network analyses are a starting point for future exploration. For example, our finding that Latine-identified patients with AD have more interconnected, centralized comorbidities can be investigated further to assess, for instance, its generalizability as well as the effect of social determinants of health on these findings.

We now include in our results the prevalence of specific phenotype pairs for patients with AD. We also included the prevalence of all phenotypes (5% cutoff) and phenotype pairs (5%) for each identified R&E in supplementary data files. The phenotype supplementary data files (supplementary data 2 and supplementary data 4) also contain information about node-level

network metrics. Details of these supplementary data files are shared below. Below is a table specifying what is contained in each supplementary data file:

Supplementary Data Containing Network Phenotype and Phenotype Pair Prevalence

Supplementary Data Number	Data File Information
2	UCSF nodes (phenotypes)
3	UCSF edges (phenotype pairs)
4	UC Validation nodes (phenotypes)
5	UC Validation edges (phenotype pairs)

We have also transferred the summary from the discussion to the results as suggested by reviewer 1.

4. It is well established that there are vast differences in utilization patterns by race, as racialized minorities are less likely to seek regular care and will have less comprehensive EHR data. This can introduce bias in EHR-based research, which must be addressed. Additionally, what is the timeframe in which the authors identified comorbidities? If some patients have been seen for longer in the healthcare system (those who seek care and/or are sicker), they will have more data than those that do not, and that can introduce bias. This becomes even more of an issue when you select controls. How can you be sure the controls really do not have Alzheimer's?

Thank you for highlighting these essential considerations.

Hospital Utilization

First, we would like to confirm that yes, we do see differences in utilization patterns when it comes to the types of visits between racialized populations in our study; for example, the figure on the next page shows differences in visit types for patients UC-wide. However, the control patients for each comparison are identified as belonging to the same racialized group.

We also looked at the number of visits to explore whether there were differences between racialized populations as another measure of hospital utilization. When looking at the number of visits for each racialized population with AD both at UCSF and in the UC-wide validation cohort, we found that the number of visits for patients with AD were at least double that of candidate control patients on average. The results are shown in tables below:

Number of visits at UCSF for each identified R&E, rounded to the nearest whole number (last visit date - first visit date)

Identified R&E	Patients with AD	Control Patients
Asian	226	112
Black	194	70
Latine	221	73
White	113	59

Number of visits at the 4 UCs for each identified R&E, rounded to the nearest whole number (last visit date - first visit date)

Identified R&E	Patients with AD	Control Patients
Asian	85	42
Black	116	51
Latine	92	37
White	92	45

Based on these findings, we acknowledge that the differences we see between racialized populations, patients with AD relative to matched controls when stratified by identified R&E, and between UCSF and the UC-wide validation cohort, could be due in part to differences in the overall hospital utilization for each population.

Length of time in the EMR

We included all diagnoses for each patient, regardless of when it was received. Therefore, the time frame considered included the complete length of time the patient was in the EMR. How long the patients have been in the EMR cannot be answered conclusively since our databases do not include visit information for all patients. This includes patients with AD at UCSF. However, we did explore the length of time patients were in the EMR with the data available.

To address whether there were differences in the length of time patients have been in the EMR, we analyzed the length of time between the first visit and last visit, when available, for the AD and control cohorts at UCSF (i.e., the cohorts in study). Since we no longer have access to an archived database for UCSF, we set January 7, 2021 to be the last visit date cutoff for all patients who have a last visit date recorded after this date. January 7, 2021 was the latest date included when the database was queried at the end of October 2021.

We also explored the length of time patients have been in the EMR for the current AD cohort at the 4 UCs included in the UC-wide validation analyses. Similar to UCSF, we did not have access to archived versions of the University of California Data Discovery Platform (UCDDP) database.

Overall, patients with AD have been in the EMR for a longer length of time relative to control (or candidate control in the case of UC-wide) patients. Below is a table showing these differences at UCSF (note, these are the patients included in analyses after two rounds of propensity score matching):

Length of time in the EMR at UCSF for each identified R&E (last visit date - first visit date)

Identified R&E	Patients with AD (years)	Control Patients (years)
Asian	13.65	9.01
Black	17.61	10.67
Latine	14.88	8.47
White	11.96	8.31

We also see these differences for patients in the current AD cohort across the 4 UCs. Since we did not have access to the archived database, we measured the length of time in the EMR for the current patients with AD (note, the current AD cohort across the 4 UCs should include most

if not all of the patients identified in the study when the database was queried in September 2021). We also measured the length of time in the EMR for candidate control patients across the 4 UCs included in the study (that is, patients 65 years or age or older without an AD diagnosis). Note that UCDDP's records begin in 2012, while UCSF's records begin in 1982, so there will be differences in the length of time patients have been in the EMR between these two databases.

Candidate control patients' length of time in the EMR across the 4 UCs is less than half of that for patients with AD (note that these are not matched control patients; since control patients were matched on death status along with other factors, these candidate control patients may be healthier relative to the matched controls in the UC-wide validation cohort).

Length of time in the EMR across the 4 UCs for each identified R&E (last visit date - first visit date)

Identified R&E	Patients with AD (years)	Control Patients (years)
Asian	7.41	3.02
Black	8.05	3.11
Latine	7.27	2.54
White	7.76	3.24

These considerations - differences in hospital utilization and length of time in the EMR - are both incredibly important and are highlighted in the discussion section on page 26: "Additionally, we did not control for the number of visits nor the length of time patients have been in the EMR, which could have an effect on the number of diagnoses a patient has and thus is a limitation to consider when interpreting findings." Understanding how these factors play a role in the context of comparing comorbidities as well as in predictive modeling are currently being explored in ongoing studies by our team.

Finally, we cannot be sure that the controls do not have AD. This highlights a fundamental limitation of EMRs; we only have the observations that we have, without access to a 'ground truth' of what the patients actually have and are experiencing. Thank you for bringing attention to this important consideration. We have now included this as a limitation in the discussion on page 26: "we cannot know for certain whether controls do not have AD, due to the fundamental inaccessibility to the ground truth of what conditions patients actually have."

5. The researchers need to indicate the representativeness by race and gender of the dataset they derived their subsets from. They created equal-sized cohorts, but it is not clear how representative the underlying population is.

Thank you for the question. Please find figures below.

Identified R&E at UCSF

Below is a pie chart depicting the overall demographic breakdown of the AD cohort in the study by identified R&E at UCSF (Supplementary Figure 1):

And below is a pie chart depicting the overall demographic breakdown of all current patients by identified R&E at UCSF:

Identified R&E across the 4 UCs (UCD, UCI, UCLA, UCSD)

Below are pie charts depicting the overall demographic breakdown of all patients, as well as the AD cohort in the study specifically, at each of the 4 UCs by identified R&E (demographics for patients with AD by identified R&E across the 4 UCs are included in Supplementary Figure 2).

Identified gender at UCSF

Below is a pie chart depicting the overall demographic breakdown of the AD cohort in the study by identified gender at UCSF:

And below is a pie chart depicting the overall demographic breakdown of all current patients by identified gender at UCSF:

Identified gender across the 4 UCs (UCD, UCI, UCLA, UCSD)

Below are pie charts depicting the overall demographic breakdown of all patients, as well as AD cohort in the study specifically, at each of the 4 UCs by identified gender.

As expected at both UCSF and across the 4 UCs, female-identified patients comprise a higher percentage of patients with AD relative to the percentage of female-identified patients overall, while male-identified patients comprised a lower percentage of patients with AD relative to the percentage of male-identified patients overall.

Reviewer #3 (Remarks to the Author):

This paper utilizes EMRs from the patients receiving care at UCSF to assess whether comorbidities differed by race (Asian, Non-Hispanic Black, Latino, Non-Hispanic white). The analyses were then replicated using patients receiving care at the larger University of California system (UC Davis, UC Irvine, UCLA, and UCSD). The authors report that although AD-associated comorbidities are generally similar across racialized populations, there are some differences. The paper is interesting and a potential starting point. However, there are several aspects that are not addressed, and the overall significance is difficult to interpret.

We would like to thank the reviewer for the summary of our work and recognizing its value. Please find the point-by-point responses below addressing the concerns.

1. Please provide an overview of which patients are seen at each of the UC sites. Are the patients seen at these sites representative of the surrounding population? Are they of higher SES and therefore not as generalizable? Are patients on Medicaid seen? All of these factors are important in assessing racial/ethnic differences due to differences in access to care.

Thank you for bringing these considerations up. Which patients are seen at each of the UC sites can be found in response to reviewer 2, question 5, under the sections *Identified R&E at UCSF* on page 9 and *Identified R&E across the 4 UCs (UCD, UCI, UCLA, UCSD)* on page 10.

Thank you for asking whether the patients seen at these sites are representative of the surrounding population; this is an essential consideration. While we are also interested in knowing whether the patients seen at these sites are representative of the surrounding population, we are limited by the information provided in the de-identified data. Within each UC site, the patients can be seen at numerous health centers. We do not know where these health centers are, nor do we know where the patients are coming from. Therefore, we do not know the answer to this question, nor can we know whether the patients in the EMR are of higher SES relative to the surrounding population.

It is also critical to know what the insurance status of our patients are and whether our cohorts include patients on Medicaid. However, this is not currently available in the OMOP databases at UCSF or UC-wide. This is a major limitation that we hope will be addressed soon.

We agree that these are all important factors when it comes to inequities in access to care, which are likely underlying the differences we see between racialized populations. Not being able to answer these questions highlights a current limitation of many EMRs; identifying these social determinants of health (SDoH), such as access to insurance coverage, as well as direct measures of exposure to different forms of racism, are not systematically recorded. We hope to work with teams based at UCSF and across UC who manage EMRs to properly include SDoH systematically in the near future; this is mentioned in the discussion section on page 25: “We plan to identify strategies to properly incorporate SDoH in future work.”

2. African Americans are diagnosed at later stages of dementia than whites (Lennon et al. 2021). Thus, the diagnosis depends on race. This makes it difficult to then compare comorbidities by race. Can the authors better respond to this?

We thank the reviewer for this question; this is indeed an important consideration. Our work is likely capturing the context in which patients are getting their diagnoses and not underlying biological or clinical factors. There are disparities in diagnosis prevalence as well as when patients are being diagnosed in racially marginalized individuals. This is a limitation of our data, but something we hope to explore in future work.

We added this limitation to the discussion section on page 25 and included the Lennon et al. study as a reference: “There may also be differences in AD diagnosis prevalence, as well as when AD is diagnosed, between individuals based on identified R&E, which has indeed been

suggested between Black- and White- identified individuals with AD (Lennon, J. C. *et al.* Black and White individuals differ in dementia prevalence, risk factors, and symptomatic presentation. *Alzheimers Dement.* **18**, 1461–1471 (2022)). We attempted to account for this by comparing diagnoses between patients with AD with matched controls, stratified by identified R&E, but there are caveats to consider to this approach as well, such as the lack of direct measures of exposure to racism, which likely differ between individuals within each identified R&E (Lett, E., Asabor, E., Beltrán, S., Cannon, A. M. & Arah, O. A. Conceptualizing, Contextualizing, and Operationalizing Race in Quantitative Health Sciences Research. *Ann. Fam. Med.* **20**, 157–163 (2022)).”

3. Similarly, there are well-known racial/ethnic differences in the diagnoses of other conditions – how might this play a role in the results?

We thank the reviewer for raising this issue. Matching patients with AD to control patients by identified R&E likely enables us to partially account for differences based on identified R&E. However, there are critical caveats to consider, such as the lack of access to direct measures of exposure to racism, which likely impact which diagnoses individuals receive within each identified R&E.

4. Can any information be obtained on education, income, or other social determinants of health?

Unfortunately, the UCSF and UCDDP OMOP-based de-identified databases currently have very limited social determinants of health information. This is an essential limitation of de-identified EMRs generally. We hope that these social determinants of health will be ethically and systematically included in the future.

5. In the discussion, the authors briefly mention that different cultures can present symptoms differently. This is an extremely important point and deserves more attention.

We would like to thank the reviewer for recognizing the importance of this consideration. We have expanded on this point in the discussion section on page 23 to add to its significance, and we also referenced four additional articles to provide more context for the reader: “Differences in symptom expression have been explored in numerous studies in the context of neuropsychiatric conditions such as depression, anxiety, and obsessive-compulsive disorder, where symptom presentations often differ based on an individual’s cultural and geographic context.”

6. I am a bit confused on the matching by death status. Why not just include individuals living at the time – which I think was between Sept and Oct 2021?

Thank you for this question. We included deceased patients to increase statistical power, since they comprise a large proportion of patients with an AD diagnosis in the EMRs. This is especially the case in the UC-wide validation cohort, where approximately 43% of the patients with AD across the 4 UCs are known to be deceased. We also wanted to account for death

rates between cases and controls, to ensure that there are not greater rates of death in controls versus cases.

7. Codes were included before and after the AD diagnoses – how many years before were codes included?

Thank you for the question. Longitudinal analysis will be a part of our future work. For this study, codes were included between approximately 6.8 years and 8.4 years before an AD diagnosis for the UCSF cohort; this is now included in the methods section on page 6: “Codes were included between approximately 6.8 years and 8.4 years before an AD diagnosis for patients with AD.” The averages are shown below:

Mean length of time between first recorded diagnosis and first AD diagnosis at UCSF for each identified R&E

Identified R&E	Patients with AD (years)
Asian	8.4
Black	8.4
Latine	8.4
White	6.8

In the validation cohort, codes were included between approximately 2.1 years and 2.3 years before an AD diagnosis for patients with AD that are currently included in the database; this is now included in the methods section on page 8: “All ICD-9-CM and ICD-10-CM codes for patients were included, including codes between approximately 2.1 years and 2.3 years before an AD diagnosis for patients with AD.” These patients should comprise patients with AD in the study in addition to patients with AD that have been added from subsequent refreshes of the UCDDP. The averages are shown below:

Mean length of time between first recorded diagnosis and first AD diagnosis across the 4 UCs for each identified R&E

Identified R&E	Patients with AD (years)
Asian	2.1
Black	2.3
Latine	2.1
White	2.3

8. Was there a race/ethnic differences in number of medical visits? Presumably the more visits the more diagnoses.

Thank you for raising this question. Please refer to our answer to reviewer #2, question 4, which addresses differences in the number of medical visits. The most salient finding was that the largest differences found were between patients with AD and control patients, which may contribute to more diagnoses found amongst patients with AD relative to control patients. There were also some differences between patients in the number of medical visits when patients were stratified by identified R&E, though these differences were more subtle. Moreover, these differences were not consistent between the two cohorts in the study. The findings are summarized in the next paragraph.

In the UCSF cohort, White-identified patients with AD had significantly fewer visits relative to Asian-, Black-, and Latine- identified patients with AD, while Asian-identified control patients had significantly more visits relative to Black-, Latine-, or White- identified control patients (Kruskal-Wallis test, followed by Dunn's test). In the most recently queried AD cohort across the 4 UCs (UCD, UCI, UCLA, UCSD), Black-identified patients had significantly more visits relative to Asian-, Latine-, or White- identified patients with AD (Kruskal-Wallis test, followed by two-sided Dunn's test). Significant differences were also found between the majority of pairwise comparisons of patients aged 65 and older when stratified by R&E category, though this may be due in part to large sample size (there were approximately 1.4 million patients compared).

REVIEWERS' COMMENTS:

Reviewer #1 (Remarks to the Author):

The authors did a wonderful job addressing all of my concerns. I have no further critiques.

Reviewer #2 (Remarks to the Author):

I would like to thank the authors for addressing my concerns very robustly. The additional analysis and updates to the paper have addressed each of my concerns. I look forward to seeing this paper in print.

Reviewer #3 (Remarks to the Author):

The authors have generally addressed my comments. I think it would be very helpful to incorporate the figures of the characteristics of patients seen at the different sites - even as supplementary figures - and not just in the response to review.

In addition, several limitations were acknowledged in response to the comments, but not added to the paper (e.g., lack of info on Medicaid). These should be added.

REVIEWERS' COMMENTS:

Reviewer #1 (Remarks to the Author):

The authors did a wonderful job addressing all of my concerns. I have no further critiques.

Reviewer #2 (Remarks to the Author):

I would like to thank the authors for addressing my concerns very robustly. The additional analysis and updates to the paper have addressed each of my concerns. I look forward to seeing this paper in print.

Reviewer #1 and Reviewer #2: Thank you both for taking the time to review our manuscript and for offering helpful advice and suggestions!

Reviewer #3 (Remarks to the Author):

The authors have generally addressed my comments. I think it would be very helpful to incorporate the figures of the characteristics of patients seen at the different sites - even as supplementary figures - and not just in the response to review.

Thank you for reviewing our manuscript and for offering additional advice. We have incorporated figures showing the breakdown of patients' identified race and ethnicities seen at different sites as supplementary figures.

In addition, several limitations were acknowledged in response to the comments, but not added to the paper (e.g., lack of info on Medicaid). These should be added.

Thank you for highlighting the importance of these limitations. To address the unavailability of insurance status, as well as the lack of availability of other SDoH indicators that could address whether patients seen at the UC health centers differ relative to the surrounding population, we included the following on page 26 of the discussion section: "Also, indicators of SDoH that likely contributed to differences we do see between racialized populations, such as insurance and Medicaid status, as well as how patients' SDoH compare to those living in the surrounding area in which they received care, were not analyzed. This is due to the current unavailability of SDoH indicators in the de-identified OMOP databases we used for this study." Additionally, to address the limitation differences in the diagnoses of other conditions between racialized populations, we included the following on page 26 of the discussion section: "Furthermore, there are differences in the prevalence of other diseases as well between racialized populations, which we also attempted to account for by performing a stratified analysis for each identified R&E (Peterson, R. L. et al. Racial/Ethnic Disparities in Young Adulthood and Midlife Cardiovascular Risk Factors and Late-life Cognitive Domains: The Kaiser Healthy Aging and Diverse Life Experiences (KHANDLE) Study. *Alzheimer Dis. Assoc. Disord.* 35, 99–105 (2021).)"